# Non-KREEP origin for Chang'e-5 basalts in the Procellarum KREEP Terrane

Heng-Ci Tian[1,4], Hao Wang[2,4], Yi Chen[2,4], Wei Yang[1✉], Qin Zhou[3], Chi Zhang[1], Hong-Lei Lin[1], Chao Huang[2], Shi-Tou Wu[2], Li-Hui Jia[2], Lei Xu[2], Di Zhang[2], Xiao-Guang Li[2], Rui Chang[1], Yue-Heng Yang[2], Lie-Wen Xie[2], Dan-Ping Zhang[2], Guang-Liang Zhang[3], Sai-Hong Yang[3] & Fu-Yuan Wu[2]

Mare volcanics on the Moon are the key record of thermo-chemical evolution throughout most of lunar history[1–3]. Young mare basalts—mainly distributed in a region rich in potassium, rare-earth elements and phosphorus (KREEP) in Oceanus Procellarum, called the Procellarum KREEP Terrane (PKT)[4]—were thought to be formed from KREEP-rich sources at depth[5–7]. However, this hypothesis has not been tested with young basalts from the PKT. Here we present a petrological and geochemical study of the basalt clasts from the PKT returned by the Chang'e-5 mission[8]. These two-billion-year-old basalts are the youngest lunar samples reported so far[9]. Bulk rock compositions have moderate titanium and high iron contents with KREEP-like rare-earth-element and high thorium concentrations. However, strontium–neodymium isotopes indicate that these basalts were derived from a non-KREEP mantle source. To produce the high abundances of rare-earth elements and thorium, low-degree partial melting and extensive fractional crystallization are required. Our results indicate that the KREEP association may not be a prerequisite for young mare volcanism. Absolving the need to invoke heat-producing elements in their source implies a more sustained cooling history of the lunar interior to generate the Moon's youngest melts.

On 17 December 2020, China's Chang'e-5 mission returned about 1.73 kg of lunar materials from one of the youngest basalt units in northern Oceanus Procellarum[8,10]. The samples studied here include two epoxy mounts, each containing two basalt clasts and two soils allocated by the China National Space Administration (Fig. 1, Extended Data Table 1). The basalt clasts in epoxy mounts and the soils were scooped from the lunar surface. The lithic clasts (larger than 0.6 mm) were picked from the two soil samples. Approximately 45% of lithic clasts are basalt. The basalt clasts show a range of textures from porphyritic to subophitic, poikilitic and equigranular (Extended Data Fig. 1). Eighteen representative basalt clasts with various textures were selected for petrological and geochemical analyses (Extended Data Table 1). The basalt clasts analysed are composed of clinopyroxene, plagioclase, olivine and ilmenite, as well as minor amounts of K-feldspar, silica, spinel, apatite, baddeleyite, zirconolite, tranquillityite and merrillite (Fig. 2a, Extended Data Table 2). Uranium (U)-rich minerals (baddeleyite, zirconolite and tranquillityite) of 13 clasts (Extended Data Table 1) were dated by the in situ lead (Pb)–Pb method, which suggested a crystallization age of 2,030 ± 4 million years ago (Ma)[9].

We analysed major and trace elements and strontium–neodymium (Sr–Nd) isotopes for different minerals in the basalt clasts. Our results show that most pyroxene and olivine grains have a low magnesium number (Mg#) (Supplementary Table 1, Fig. 2) and generally show compositional zoning with Mg-rich cores and iron-rich rims (Extended Data Fig. 2). The chemical compositions of pyroxene indicate crystallization temperatures ranging from 1,200 °C to 1,000 °C for the cores and below 800 °C for the rims (Fig. 2c). Plagioclase and K-feldspar in all clasts are homogeneous from the core to the rim within the grains, but show considerable compositional variations among different grains (anorthite (An)$_{76–90}$ and orthoclase (Or)$_{62–93}$; Extended Data Fig. 3a). Pyroxene shows parallel rare-earth element (REE) patterns with a large variation from 10 to 50 times that of carbonaceous chondrites. (Supplementary Table 2, Extended Data Fig. 4). Fourteen plagioclase and five merrillite grains in the clasts yield homogeneous and low initial $^{87}Sr/^{86}Sr$ ratios of 0.69934 to 0.69986 and positive $\varepsilon_{Nd}(t)$ values of 7.9 to 9.3, respectively (Extended Data Tables 3, 4). $\varepsilon_{Nd}(t) = ((^{143}Nd/^{144}Nd)$ sample$(t)/(^{143}Nd/^{144}Nd)_{CHUR} − 1) \times 10,000$, where $^{143}Nd/^{144}Nd)$sample$(t)$ and $^{143}Nd/^{144}Nd)_{CHUR}$ are the Nd isotopic compositions of sample and Chondritic Uniform Reservoir (CHUR) at time ($t = 2,030$ Ma), respectively. Both the calculated $^{147}Sm/^{144}Nd$ and $^{87}Rb/^{86}Sr$ of the source do not lie on the array of potassium, rare-earth elements and phosphorus (KREEP)-rich material, but rather indicate a light-REE (LREE)-depleted mantle source (Fig. 3).

Although the studied clasts have a range of petrographic textures and modal abundances of minerals (Extended Data Table 2), three lines of evidence suggest that they are most likely from a single basaltic

[1]Key Laboratory of Earth and Planetary Physics, Institute of Geology and Geophysics, Chinese Academy of Sciences, Beijing, China. [2]State Key Laboratory of Lithospheric Evolution, Institute of Geology and Geophysics, Chinese Academy of Sciences, Beijing, China. [3]National Astronomical Observatories, Chinese Academy of Sciences, Beijing, China. [4]These authors contributed equally: Heng-Ci Tian, Hao Wang, Yi Chen. ✉e-mail: yangw@mail.iggcas.ac.cn

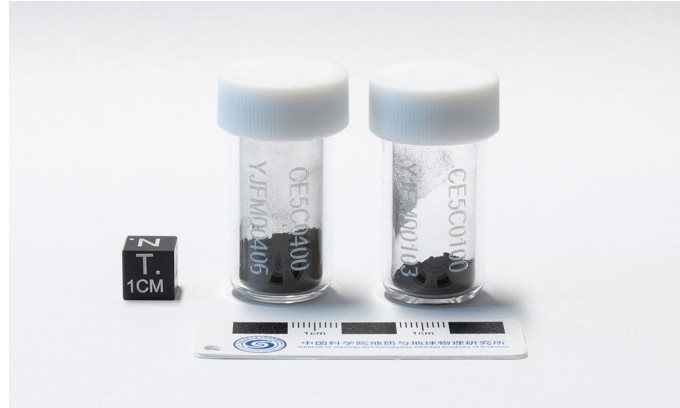

**Fig. 1 | Chang'e-5 soil samples.** Photo of soil samples CE5C0100YJFM00103 and CE5C0400YJFM00406. Photo taken by Hui Ren.

lava flow. First, the initial plagioclase $^{87}Sr/^{86}Sr$ ratios of 11 clasts and the merrillite $\varepsilon_{Nd}(t)$ of three clasts are nearly identical (Fig. 3). Second, most of the pyroxene grains in these clasts fall along the 1:2 line on a titanium/aluminium (Ti/Al) diagram (Extended Data Fig. 3b), reflecting the near-simultaneous crystallization of augite and plagioclase during one cooling event[11]. Lastly, both the pyroxene and plagioclase grains measured in different clasts show similar REE patterns (Extended Data

Fig. 4), like those in the Apollo 12 and Apollo 15 basalts with different cooling rates[12]. Therefore, the range in textures of the Chang'e-5 basalt clasts is probably due to different cooling rates[13] within different parts of the lava flow. The appreciable variation of mineralogical abundances and inferred whole-rock compositions are probably due to the small sample sizes (smaller than 3 mm) of the basalt clasts.

For comparison with previous results from the Apollo and Luna samples and remote sensing, the bulk-rock major and trace-element compositions of the Chang'e-5 basalts are estimated (Extended Data Tables 5, 6). The Chang'e-5 basalt clasts can be classified as low-Ti/high-Al/low-potassium (K) type[14]. However, these basalts have higher iron (FeO; 22.2 wt%), titanium (TiO$_2$; 5.7 wt%) and aluminium (Al$_2$O$_3$; 11.6 wt%) contents and a lower Mg# (32.1) relative to the Apollo and Luna low-Ti basalts (Fig. 2b, Extended Data Fig. 5). In addition, the Chang'e-5 clasts are highly LREE enriched (50 times heavy REE (HREE) and 150 times LREE enrichment relative to carbonaceous chondrites; Fig. 4a), and show high thorium (Th) contents (approximately 4.5 ppm; Extended Data Table 6). The REE patterns are different from those of the Apollo low-Ti basalts, but parallel to those of the KREEP basalts (Fig. 4a). In addition, the FeO, TiO$_2$ and Th contents of the Chang'e-5 basalt agree well with the data obtained by the Lunar Prospector Gamma-Ray Spectrometer[15–17] (FeO about 22.4 wt%; TiO$_2$ about 4.5 wt%; Th about 5.8 ppm). This agreement suggests that the Chang'e-5 basalt clasts are representative of the mare basalt unit of the landing site (designated as unit P58; ref.[18]).

The elevated incompatible trace element (ITE) concentrations and LREE enrichment in the Chang'e-5 basalt clasts are typical characteristics

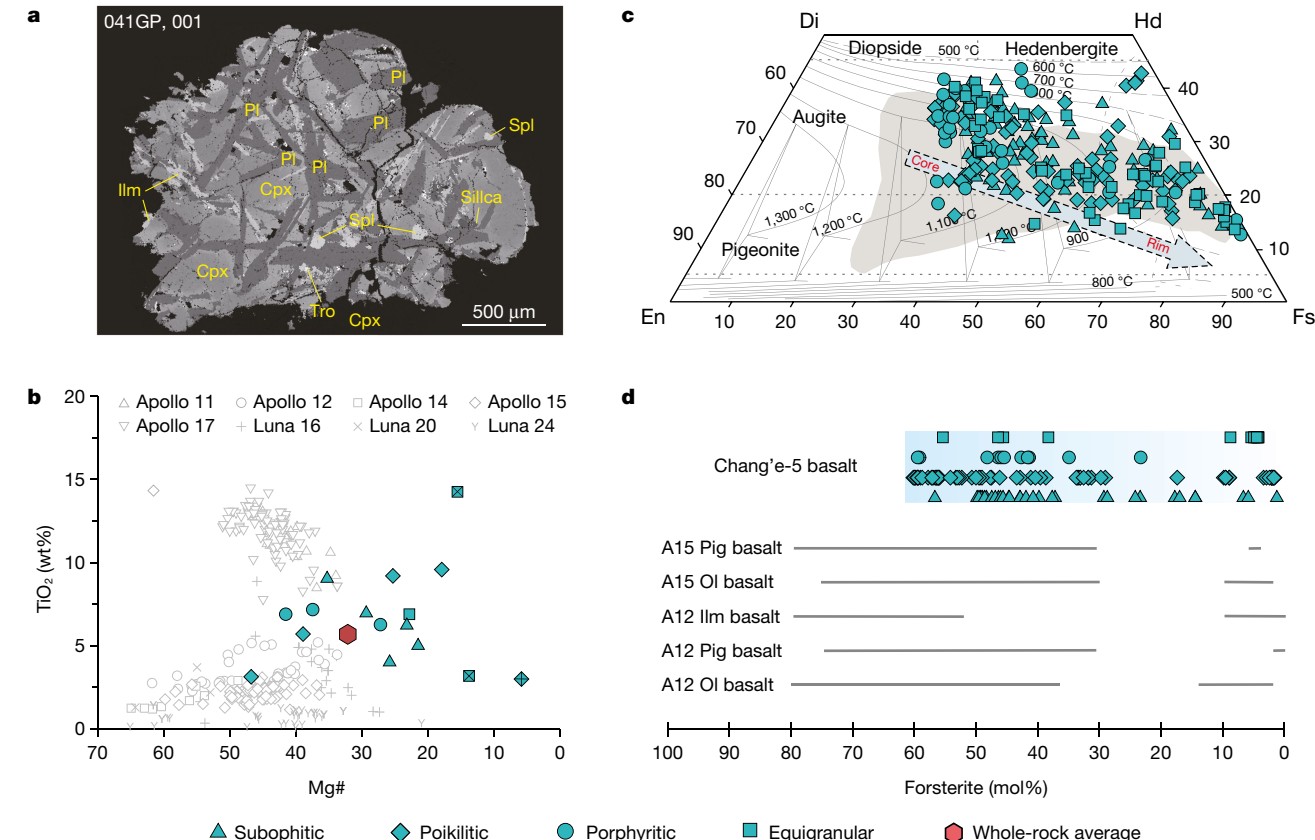

**Fig. 2 | Petrography and mineral chemistry of the Chang'e-5 basalt clasts.** **a**, Backscattered electron image of Chang'e-5 basalt clast 041GP, 001. Cpx, clinopyroxene; Ilm, ilmenite; Pig, pigeonite; Pl, plagioclase; Ol, olivine; Spl, Spinel; Tro, troilite. **b**, Mg# versus TiO$_2$ diagram of the Chang'e-5 basalt clasts. The Apollo and Luna data are from Clive Neal's Mare Basalt Database (https://www3.nd.edu/~cneal/Lunar-L/). **c**, Quadrilateral diagram of pyroxene in the Chang'e-5 basalt clasts. Temperature contours[33] calculated at 0.5 GPa are shown, where 0.5 GPa was chosen according to the possible pressure range for lunar low-Ti basalts[34]. The Apollo 12 samples[35] are plotted (grey area) for comparison. Di, diopside; En, enstatite; Fs, ferrosilite; Hd, hedenbergite. **d**, Comparison of olivine compositions in Chang'e-5 basalts with those within the Apollo 12 and Apollo 15 basalts[36]. The data for pyroxene and olivine are provided in Supplementary Table 1.

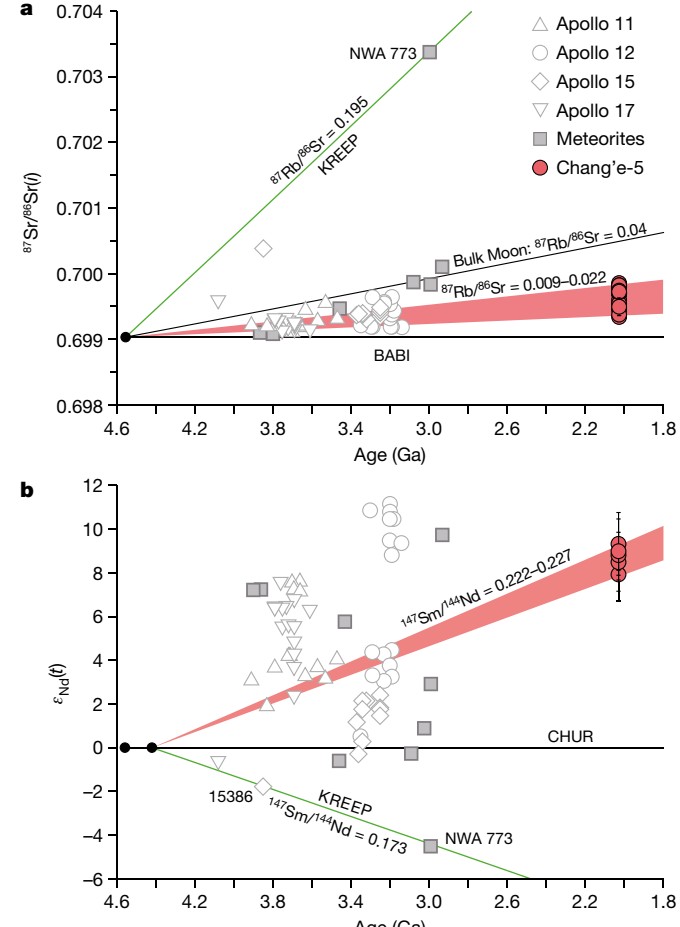

**Fig. 3 | Rb–Sr and Sm–Nd isotopic evolution of lunar materials. a**, $^{87}Rb/^{86}Sr$ ratios of Chang'e-5 basalt source regions are calculated assuming a single-stage model in which the Moon differentiated at 4.56 Ga with an initial $^{87}Sr/^{86}Sr$ ratio of 0.69903 (refs. [37,38]). The bulk Moon $^{87}Rb/^{86}Sr$ value is from refs. [38,39]. **b**, The $^{147}Sm/^{144}Nd$ ratios of the basalt source regions are calculated assuming a two-stage growth model following refs. [20,21]. In this model, the Moon followed a chondritic path until differentiation occurred at 4.42 ± 0.07 Ga, represented by the model age of primeval KREEP formation[40,41]. The Sr and Nd isotopic data of the Chang'e-5 basalts were acquired on plagioclase and merrillite, respectively (Extended Data Tables 3, 4). The initial $^{87}Sr/^{86}Sr$ and $\varepsilon_{Nd}(t)$ are calculated using 2.03 Ga (ref. [9]). The horizontal solid lines in both panels refer to the primordial reservoir. BABI, Basaltic Achondrite Best Initial; CHUR, Chondrite Uniform Reservoir. The Apollo mare basalts and meteorites data are from ref. [21] and references therein.

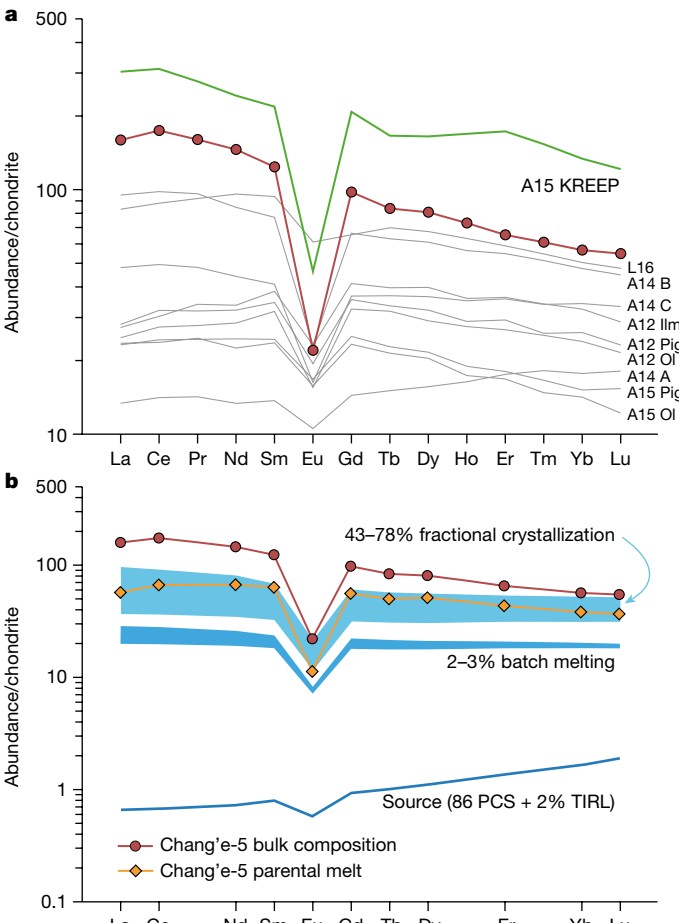

**Fig. 4 | Formation of the REE distribution patterns in Chang'e-5 basalts. a**, Comparison of REE distribution patterns of Chang'e-5 and Apollo basalts. Apollo (A) and Luna (L) data are from Clive Neal's Mare Basalt Database (https://www3.nd.edu/~cneal/Lunar-L/). The Apollo 14 groups A, B and C were defined by ref. [42]. **b**, REE modelling of partial melting and fractional crystallization. The blue areas denote the melts produced after 2–3% partial melting of the mantle source (86 PCS + 2% TIRL[22]; PCS, per cent crystallized solid; TIRL, trapped instantaneous residual liquid) and those followed by 43–78% fractional crystallization. This source composition is calculated based on the source region $^{87}Rb/^{86}Sr$ and $^{147}Sm/^{144}Nd$ ratios (Fig. 3). Mineral modes in the source are assumed to be 48% olivine, 23% orthopyroxene, 23% pigeonite, 3% augite and 3% plagioclase. Mineral assemblages of 43–78% crystallization are: 5–10% olivine, 25–59% augite, 2–3% pigeonite and 6–11% plagioclase. Normalization values are from ref. [43]. The Chang'e-5 parental melt estimated by the clinopyroxene core with the highest Mg# (sample 406-004, 005) (Extended Data Table 6) requires extensive (43–78%) fractional crystallization after low-degree (2–3%) melting of the mantle source. To match the Chang'e-5 bulk composition, up to 78–88% fractional crystallization is needed. The model parameters are listed in Supplementary Table 3, and details of the batch melting and fractional crystallization model are provided in the Methods.

for the KREEP-rich materials. However, the isotopic compositions of the Chang'e-5 basalt are not consistent with the origin of KREEP-rich materials (Fig. 3). Even a small contribution (less than 0.5%; Extended Data Fig. 6) from KREEP materials would result in high $^{87}Rb/^{86}Sr$ ratios (greater than 0.19) and low $^{147}Sm/^{144}Nd$ ratios (less than 0.173) (Fig. 3), which would shift the Sr–Nd isotopes of the Chang'e-5 basalts considerably. The low initial $^{87}Sr/^{86}Sr$ and high $\varepsilon_{Nd}(t)$ observed in the Chang'e-5 basalts are similar to those of the Apollo 12 low-Ti basalts. This similarity indicates that both the Chang'e-5 basalts and the Apollo 12 low-Ti basalts may originate from a depleted non-KREEP source, which may have crystallized from the early lunar magma ocean cumulates dominated by olivine and pyroxene[19].

The REE and ITE enrichment most likely formed through magmatic processes, such as partial melting and fractional crystallization. Similar characteristics observed in lunar basalt meteorites (for example, NWA 032, NWA 4734 and LAP 02205) were considered to originate

from low-degree partial melting of a depleted source[20,21]. We suggest that a cumulate composition with a crystallized percent solid at 86% and the addition of 2% trapped instantaneous residual liquid in the model of ref. [22] can produce a source with $^{147}Sm/^{144}Nd \approx 0.222–0.227$ and $^{87}Rb/^{86}Sr \approx 0.009–0.022$ (Fig. 2). Partial melting of such a mantle source alone cannot simultaneously reproduce the LREE and HREE contents even with an unrealistic, low degree of melting (less than 0.3%) (Extended Data Fig. 7). Thus, fractional crystallization must have occurred to elevate the ITE and LREE abundances before the eruption of the Chang'e-5 basalts. Even the parental melt estimated by the clinopyroxene core with the highest Mg# (sample 406-004, 005) (Extended Data Table 6) still

**Table 1 | Summary of age and chemical compositions of the Apollo and Chang'e-5 basalts**

| Name | Chang'e-5 | A12 Ol | A12 Pig | A12 Ilm | A15 Ol | A15 Pig | A15 KREEP |
|---|---|---|---|---|---|---|---|
| Age (Ga) | 2.03 | ~3.15–3.17 | ~3.13–3.18 | ~3.18–3.19 | ~3.27–3.29 | ~3.35–3.37 | 3.89 |
| FeO (wt%) | 22.2 | 20.6 | 21.0 | 19.9 | 20.9 | 21.3 | 10 |
| $TiO_2$ (wt%) | 5.7 | 3.6 | 3.0 | 3.7 | 2.2 | 2.1 | 2 |
| $Al_2O_3$ (wt%) | 11.6 | 9.2 | 8.1 | 10.0 | 9.4 | 8.9 | 15.5 |
| $K_2O$ (wt%) | 0.1 | 0.07 | 0.06 | 0.07 | 0.08 | 0.04 | 0.6 |
| Mg# | 32.1 | 46.4 | 52.1 | 43.2 | 47.7 | 44.8 | 61.1 |
| Th (ppm) | 4.5 | 0.74 | 1.09 | 0.79 | 0.49 | 0.56 | 11.5 |
| La (ppm) | 37.5 | 5.8 | 6.6 | 6.4 | 5.5 | 5.4 | 72 |
| $(La/Sm)_N$ | 1.29 | 0.78 | 0.81 | 0.71 | 1.0 | 0.95 | 1.39 |
| $^{87}Sr/^{86}Sr(i)$ | ~0.69934–0.69986 | ~0.69949–0.69988 | ~0.69958–0.69973 | ~0.69932–0.69955 | ~0.69914–0.69930 | ~0.69923–0.69937 | 0.70038 |
| $\varepsilon_{Nd}(t)$ | ~7.9–9.3 | ~4.3–4.5 | ~4.1–5.4 | ~9.8–11.2 | ~2.2–2.4 | ~0–2.6 | −1.8 |

The major and trace element and age data of Apollo (A) basalts are from refs. [30–32]. The subscript N represents the chondrite normalized ratios. Sr and Nd isotopic data are the same as those in Fig. 3. $^{87}Sr/^{86}Sr(i)$ and $\varepsilon_{Nd}(t)$ are calculated based on the Pb/Pb age of 2.03 Ga.

requires extensive (43–78%) fractional crystallization after low-degree (2–3%) melting of the mantle source (Fig. 4). This scenario is consistent with the low-Mg# (32.1), high-FeO (22.2 wt%) and high-$TiO_2$ (5.7 wt%) signatures (Table 1), and compositional zonings in olivine and pyroxene of the samples (Extended Data Fig. 2). Therefore, the Chang'e-5 basalts are highly evolved magmatic products, which implies the presence of a huge magma chamber beneath the PKT at that time.

The youngest mare basalt units in the PKT (for example, P56, P58, P59 and P60; ref. [18]) all show moderate-$TiO_2$ (about 3.8–5.7 wt%) and high-Th (4.9–7.3 ppm) contents[15,23]. The high-Th materials excavated by impact craters indicate a KREEP layer beneath the PKT[24]. Melting for the prolonged volcanism in the PKT was thought to be driven by the high abundances of radiogenic heat-producing elements from the KREEP-rich materials[5–7]. However, we find that the elevated Th and other incompatible elements of the Chang'e-5 basalts from the mare unit P58 were not involved with KREEP-rich materials, but rather attributable to the highly evolved basalt produced by low-degree partial melting and extensive fractional crystallization. We deem it unlikely that the KREEP-rich materials provided the heat for partial melting without being involved in the melt itself[7]. Therefore, the presence of a speculative thick KREEP layer at the base of the crust to generate the young volcanism in the PKT region is not required.

The highly evolved origin of the 2 billion-year-old (Ga) Chang'e-5 basalts implies that the lunar interior was substantially cooler at that time than at around 3.5 Ga when the variety of more primitive basalts sampled by Apollo were formed. In spite of this considerable secular cooling, there must also have been some mechanisms to keep the melt zones in the lunar mantle from solidifying until after 2 Ga (ref. [2]). One possible mechanism is that a thick insulating outer layer of the Moon called megaregolith[2,25] served as a thermal lid, resulting in a sufficiently slow cooling rate. In addition, lunar cooling can thicken the lithosphere and thus inhibit surface eruption[26]. Therefore, the thinnest crust of the PKT region (typically less than 30 km; ref. [27]) could be a critical factor that facilitated the eruption of young basalts[26]. Lastly, evidence for the lunar magnetic field persisting until sometime after around 1.92 Ga (ref. [28]) is consistent with sources of lunar interior heat flow such as the latent heat of crystallization of the solid inner core[29] being sustained until the age of the Chang'e-5 basalts. Any new model for the thermal evolution of the Moon needs to fit the observation of a non-KREEP origin for the youngest basalts of the PKT.

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

## Methods

### Sample preparation

The Chang'e-5 samples studied in this work are two one-inch epoxy mounts (CE5C0000YJYX041GP, 2 mg; CE5C0000YJYX042GP, 44.6 mg) and two soil samples (CE5C0100YJFM00103, about 1,000 mg; CE5C0400YJFM00406, about 2,000 mg) allocated by the China National Space Administration. All these samples were scooped from the lunar surface. The two one-inch epoxy mounts, each containing two basalt clasts, were already polished before allocation. For soil samples, clasts were picked up by a sieve (aperture 600 μm) and hand-picked. Then, the clasts were embedded in epoxy mounts and polished using the grinder.

### Scanning electron microscopy analysis and energy dispersive spectrometry mapping

The petrography was carried out on a Zeiss Gemini 450 field emission scanning electron microscope at the Institute of Geology and Geophysics, Chinese Academy of Sciences (IGGCAS). The acceleration voltage was 15.0 KV and the probe current was 2.0 nA. In addition, a Thermo Scientific Apreo scanning electron microscope equipped with an energy dispersive spectrometer was used to identify the phosphorus/zirconium-bearing minerals and to calculate the modal abundance of each mineral based on the elemental mapping.

### Electron microprobe analysis of minerals

The major element concentrations of pyroxene, plagioclase, olivine, ilmenite, spinel, quartz, sulfide and phosphates in each sample were analysed by a JEOL JXA8100 electron probe at the IGGCAS. The conditions of the electron microprobe analysis were: acceleration voltage of 15 kV, probe current of 20 nA, focused beam and peak counting time of 10 s. The calibration of the elemental data was done using a series of natural minerals and synthetic materials. On the basis of the analysis of internal laboratory standards, the precision for major (more than 1.0 wt%) and minor (less than 1.0 wt%) elements are better than 1.5% and 5.0%, respectively.

### In situ trace-element analysis

The trace-element abundances of pyroxene and plagioclase in basalt fragments were determined by laser ablation–inductively coupled plasma–mass spectrometry (LA–ICP–MS) employing an Element XR HR–ICP–MS instrument coupled to a 193-nm argon-fluoride excimer laser system (Geolas HD) at the IGGCAS. The approach is similar to that outlined in ref. [44] with isotopes measured using a peak-hopping mode. The laser diameter is about 32 μm with the repetition rate of 3 Hz. The laser energy density is approximately 3.0 J cm$^{-2}$. The Element XR is equipped with a high-capacity interface pump (OnTool Booster 150) in combination with Jet sample and normal H-skimmer cones to achieve a detection efficiency in the range of 1.5% (based on U in a single spot ablation of NIST SRM 612). Helium was employed as the ablation gas to improve the transporting efficiency of ablated aerosols. NIST SRM 610 (ref. [45]) reference glass was used for external calibration. ARM-1 (ref. [46]), BCR-2G (ref. [47]) and BIR-1G (ref. [47]) glasses were used for quality control monitoring. The bulk normalization as 100 wt% strategy was used for data reduction, which is accomplished using the Iolite software package with an in-house-built data reduction scheme code[48]. For most trace elements (more than 0.05 ppm), the accuracy is better than ±15% with analytical precision (1 relative standard deviation) of ±10%. The spots analysed for trace elements are shown in Supplementary Figs. 1–3.

### In situ Sr–Nd isotopic analysis

In situ Sr–Nd isotopic measurements by LA–multiple collector (MC)–ICP–MS followed the method of refs. [49–51], hence only a brief description is given below. All analyses were conducted at the IGGCAS.

A Neptune Plus MC–ICP–MS coupled to an Analyte G2 193-nm argon-fluoride excimer laser ablation system was used to determine the Sr isotopic ratios of the plagioclase. A spot size of 85 μm was employed with a repetition rate of 6 Hz and an energy density of approximately 7 J cm$^{-2}$. The Sr isotopic data were acquired by static multi-collection in low-resolution mode using nine Faraday collectors. Before laser analysis, the Neptune Plus MC–ICP–MS was tuned using NBS 987 standard solution to obtain maximum sensitivity. A typical data acquisition cycle consisted of a 30 s of measurement of the krypton gas blank with the laser switched off, followed by 60 s of measurement with the laser ablating. Data reduction was conducted offline and the potential isobaric interferences were accounted for in the following order: Kr$^+$, Yb$^{2+}$, Er$^{2+}$ and Rb$^+$. Finally, the $^{87}Sr/^{86}Sr$ ratios were calculated and normalized from the interference-corrected $^{86}Sr/^{88}Sr$ ratio using an exponential law. The whole data-reduction procedure was performed using an in-house Excel VBA (Visual Basic for Applications) macro program. The JH56 in-house plagioclase reference material was measured before and after unknown samples for external calibration[49], which are shown in Extended Data Table 3.

The same LA–MC–ICP–MS system as for the in situ Sr isotopic analysis was used to determine Nd isotopic compositions of the merrillite. Before laser analyses, the Neptune Plus MC–ICP–MS was tuned and optimized for maximum sensitivity using JNdi-1 standard solution. A laser spot size of 20 μm was employed with a repetition rate of 3 Hz and an energy density of approximately 6 J cm$^{-2}$. Each spot analysis consisted of approximately 60 s of data acquisition with the laser fire on. The SDG in-house apatite reference material was measured before and after the merrillite samples.

To obtain accurate $^{147}Sm/^{144}Nd$ and $^{143}Nd/^{144}Nd$ data by LA–MC–ICP–MS, care must be taken to adequately correct for the contribution of the isobaric interference of $^{144}Sm$ on the $^{144}Nd$ signal. The (samarium) Sm interference correction is complicated by the fact that the $^{146}Nd/^{144}Nd$ ratio, which is conventionally used to normalize the other Nd isotope ratios, is also affected by Sm interference. As a result, the mass bias correction of $^{144}Sm$ interference on $^{144}Nd$ cannot be applied directly from the measured $^{146}Nd/^{144}Nd$ ratio. In this work, we adopted the recently revised Sm isotopic abundances ($^{147}Sm/^{149}Sm = 1.08680$ and $^{144}Sm/^{149}Sm = 0.22332$)[50]. First, we used the measured $^{147}Sm/^{149}Sm$ ratio to calculate the mass bias factor of Sm and the measured $^{147}Sm$ intensity by employing the natural $^{147}Sm/^{144}Sm$ ratio of 4.866559 to estimate the Sm interference on mass 144. Then the interference-corrected $^{146}Nd/^{144}Nd$ ratio can be used to calculate the mass bias factor of Nd. Finally, the $^{143}Nd/^{144}Nd$ and $^{145}Nd/^{144}Nd$ ratios were normalized using the exponential law. The $^{147}Sm/^{144}Nd$ ratio of unknown samples can also be calculated using the exponential law after correcting for isobaric interference of $^{144}Sm$ on $^{144}Nd$ as described above. The $^{147}Sm/^{144}Nd$ ratio was then externally further calibrated against the $^{147}Sm/^{144}Nd$ ratio of the SDG reference material during the analytical sessions[50]. The raw data were exported offline and the whole data-reduction procedure was performed using an in-house Excel VBA macro program. The LREE glass analysed in this session gave a mean $^{143}Nd/^{144}Nd$ ratio of $0.512100 \pm 0.000048$ (2 s.d., $n = 8$), which is consistent with the recommended value[52]. The data are shown in Extended Data Table 4. The spots analysed for Sr–Nd isotopes are shown in Supplementary Figs. 1–3.

### Petrography and mineral chemistry

The lithic clasts from the two lunar soils comprise about 45% basalts, about 35% impact melt breccias and about 20% agglutinates. The basalt clasts can be texturally subdivided into four types: poikilitic, subophitic, porphyritic and equigranular. Poikilitic (about 40%) and subophitic (about 40%) clasts dominate and porphyritic (about 10%) and equigranular (about 10%) clasts are minor.

The poikilitic clasts are mainly composed of clinopyroxene, plagioclase, olivine and accessory Cr-Ti-spinel, ilmenite, troilite and mesostasis including K-feldspar, fayalite, silica and phosphates. They show

various-grain-size clinopyroxene and olivine included in coarse-grained (greater than 200 μm) plagioclase (Extended Data Fig. 1a). Plagioclase is anorthite rich ($An_{76.5–89.0}$). Clinopyroxene shows a large compositional range, with Mg-rich cores (wollastonite (Wo)$_{23.8–39.4}$ enstatite (En)$_{30.2–45.8}$) and Fe-rich rims (Wo$_{15.7–42.8}$En$_{2.2–28.9}$; Supplementary Table 1). Olivine occurs as anhedral inclusions (forsterite (Fo)$_{29.7–60.1}$) in plagioclase or as mesostasis phase (fayalite, Fo$_{1.5–9.5}$). Euhedral spinel has about 10.3–19.2 wt% chromium(III) oxide ($Cr_2O_3$), about 49.3–56.5 wt% FeO and about 21.6–28.4 wt% $TiO_2$, and can be occasionally observed as inclusions in clinopyroxene and plagioclase (Extended Data Fig. 1a), pointing to an early crystallization phase.

The subophitic clasts show various grain sizes (30–300 μm) and consist mainly of plagioclase, clinopyroxene, olivine and ilmenite, with minor troilite and cristobalite (Extended Data Fig. 1b). Both clinopyroxene and olivine have compositional zoning, with Mg-rich cores and iron-rich rims (Extended Data Fig. 2, Supplementary Table 1). Plagioclase has a euhedral-to-subhedral shape with an anorthite-rich composition ($An_{75.7–90.3}$). Small amounts of Fe-rich olivine (Fo < 10) associated with cristobalite and apatite occur as mesostasis phases representing the late-stage crystallization products.

The porphyritic clasts commonly show coarse-grained (100–300 μm) mafic phenocrysts in a fine-grained (smaller than 60 μm) matrix. The mafic phenocrysts include subhedral clinopyroxene, euhedral-to-subhedral olivine and euhedral Cr-spinel (Extended Data Fig. 1c). The clinopyroxene phenocrysts are zoned from Mg-rich cores to iron-rich rims (Supplementary Table 1). The olivine phenocrysts also show compositional zoning (Fo$_{41.4–59.4}$). The Cr-spinel phenocrysts are also compositionally homogeneous, with about 24.5 wt% $Cr_2O_3$, about 2.2–2.9 wt% MgO and about 17.3–17.8 wt% $TiO_2$ contents. The matrix is composed of acicular plagioclase ($An_{76.3–85.2}$), interstitial clinopyroxene and tiny (smaller than 10 μm) Cr-spinel (Extended Data Fig. 1c). Compared with the Cr-spinel phenocrysts, the matrix ones have higher-$TiO_2$ (about 21.2–28.0 wt%) but lower-$Cr_2O_3$ contents (8.8–15.7 wt%). Ilmenite needles commonly show three directions cutting the matrix plagioclase and pyroxene, representing a late-stage crystallization phase.

The equigranular clasts are rare and show similar grain sizes (mostly about 100–200 μm) for clinopyroxene and plagioclase. Similar to the poikilitic and subophitic clasts, this type of clast contains clinopyroxene, olivine, plagioclase and ilmenite, with minor troilite and cristobalite (Extended Data Fig. 1d). Clinopyroxene shows significant compositional zoning (Wo$_{13.7–41.1}$En$_{1.1–35.9}$). The coarse-grained (larger than 100 μm) olivine shows a limited compositional range (Fo$_{35.1–43.2}$); however, the interstitial olivine grains associated with silica and apatite are systematically fayalite (Fo < 5). Plagioclase has a limited compositional range ($An_{75.5–83.1}$; Supplementary Table 1).

Except for the above four types of clast, very small amounts of coarse-grained (larger than 100 μm) fragments of basalt clasts can be found. However, their texture cannot be identified well owing to the limited (about two to five) grains in a single clast. These fragments typically contain coarse-grained clinopyroxene, plagioclase and olivine, with minor fine-grained (mostly smaller than 30 μm) silica and phosphate minerals.

### Estimation of bulk composition
It is assumed that the analysed area proportions are equal to volume proportions and the volume proportions are then converted to mass proportions based the mineral densities reported in previous studies[53–55]. The bulk composition is then calculated by their mass proportions. The oxide concentrations for the bulk composition are normalized to 100%. The average compositions of all samples are calculated based on the contribution for each sample multiplied by its weight, assuming that the weight of each sample is in proportion to its surface area (Supplementary Table 1, Extended Data Table 5). Three clasts (406-002, 002; 406-002, 007; 406-005, 010) have mineral abundances that deviate from other clasts. Clast 406-002, 002 has

an extremely high abundance of plagioclase (72.8%). Clast 406-002, 007 has a very high abundances of fayalitic olivine (22.6%) and silica (7.8%). Clast 406-002, 002 has an extremely high abundance of ilmenite (19.1%). These outlier clasts are excluded for the estimation of the bulk composition of the Chang'e-5 basalt (Extended Data Table 5).

The bulk trace elements of the Chang'e-5 basalt were estimated based on the average of all measured pyroxene grains and the partition coefficients (Extended Data Table 6). This estimation assumes that the Chang'e-5 basalt crystallized in a closed system and can be represented by the equilibrium melt of pyroxene. This method may yield large uncertainties owing to the significant variations of the trace-element contents of pyroxene. Therefore, the parental melt estimated by the clinopyroxene core with the highest Mg# (sample 406-004, 005; Extended Data Table 6) is used for the REE modelling of partial melting and fractional crystallization (Fig. 4).

### Batch melting model
In this work, we use the batch melting model to calculate the REE concentrations in the parental melts, assuming that each mineral phase melts in proportion to its modal abundance in the source. The batch melting is calculated using the following equation: $C_L/C_0 = 1/[D_0 + F(1 − D_0)]$, where $C_L$ is the weight concentration of a trace element in the melt, $C_0$ is the weight concentration of a trace element in the original cumulate source, $F$ is the weight fraction of melt produced and $D_0$ is the bulk distribution coefficient of the original solid material.

The bulk distribution coefficient is calculated by multiplying each mineral partition coefficient by the fraction of that mineral in the source. The REE partition coefficients for olivine[56], orthopyroxene[57], augite[57], pigeonite[58] and plagioclase[59], and the Th partition coefficient for augite[60] are shown in Supplementary Table 3. Given that the Chang'e-5 basalts have a similar source to the Apollo 12 basalts, the modal mineralogical assemblage calculated for Apollo 12[19] are also adopted here but with a little modification owing to the geochemical differences between Chang'e-5 and Apollo 12 mare basalts. The source materials are compiled in Extended Data Table 6.

Using the bulk distribution coefficients ($D_0$) and solid cumulate ($C_0$), the weight concentration of REE in the melt ($C_L$) is calculated for increasing melt fractions ($F$).

### Fractional crystallization model
The trace-element concentrations in the remaining melt induced by fractional crystallization are calculated using the Rayleigh fractionation equation: $C_L/C_0 = (1 − F)^{D−1}$, where $D$ is bulk distribution coefficient (the same as described in batch melting model), $F$ is the mass fraction of crystals crystallized from the melt, $C_0$ is the concentration of an element in the initial melt and $C_L$ is the concentration in the final melt. The initial melts during the calculation are assumed to be derived from 2% and 3% batch melting of the mantle source, and the results are shown in Fig. 4.

### Data availability
All data generated or analysed during this study are available in Earth-Chem Library at https://doi.org/10.26022/IEDA/112076. Source data are provided with this paper.

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

**Acknowledgements** The Chang'e-5 lunar samples were provided by the China National Space Administration. We thank R. Mitchell, H. Hui, Y. Li, X.-H. Li, Y. Lin and J. Delano for constructive comments; C. Sun for logistical support; and X. Yan and M. Liang for help with the figures. The photo in Fig. 1 was taken by Hui Ren. This study was funded by the Key Research Program of the Chinese Academy of Sciences (ZDBS-SSW-JSC007-15), the Strategic Priority Research Program of Chinese Academy of Sciences (XDB 41000000), the CAS Interdisciplinary Innovation Team, the pre-research project on Civil Aerospace Technologies of China National Space Administration (grant number D020203), and the key research programme of the Institute of Geology and Geophysics, Chinese Academy of Sciences (IGGCAS-202101). Special thanks goes to the Youth Innovation Promotion Association of Chinese Academy of Sciences for long-term support.

**Author contributions** W.Y. led the study; W.Y., Q.Z., H.-C.T., C.Z., D.-P.Z., G.-L.Z. and S.-H.Y. prepared the samples; H.-C.T., L.-H.J., D.Z., X.-G.L., S.-T.W., Y.C. and H.W. performed the major and trace-element analyses; C.H., L.X., Y.-H.Y., L.-W.X. and H.W. performed the Sr–Nd isotope analyses; H.-L.L. and R.C. contributed to data sorting and compilation; W.Y., H.-C.T., H.W., Y.C. and F.-Y.W. wrote the manuscript.

**Competing interests** The authors declare no competing interests.

**Additional information**
**Correspondence and requests for materials** should be addressed to Wei Yang.

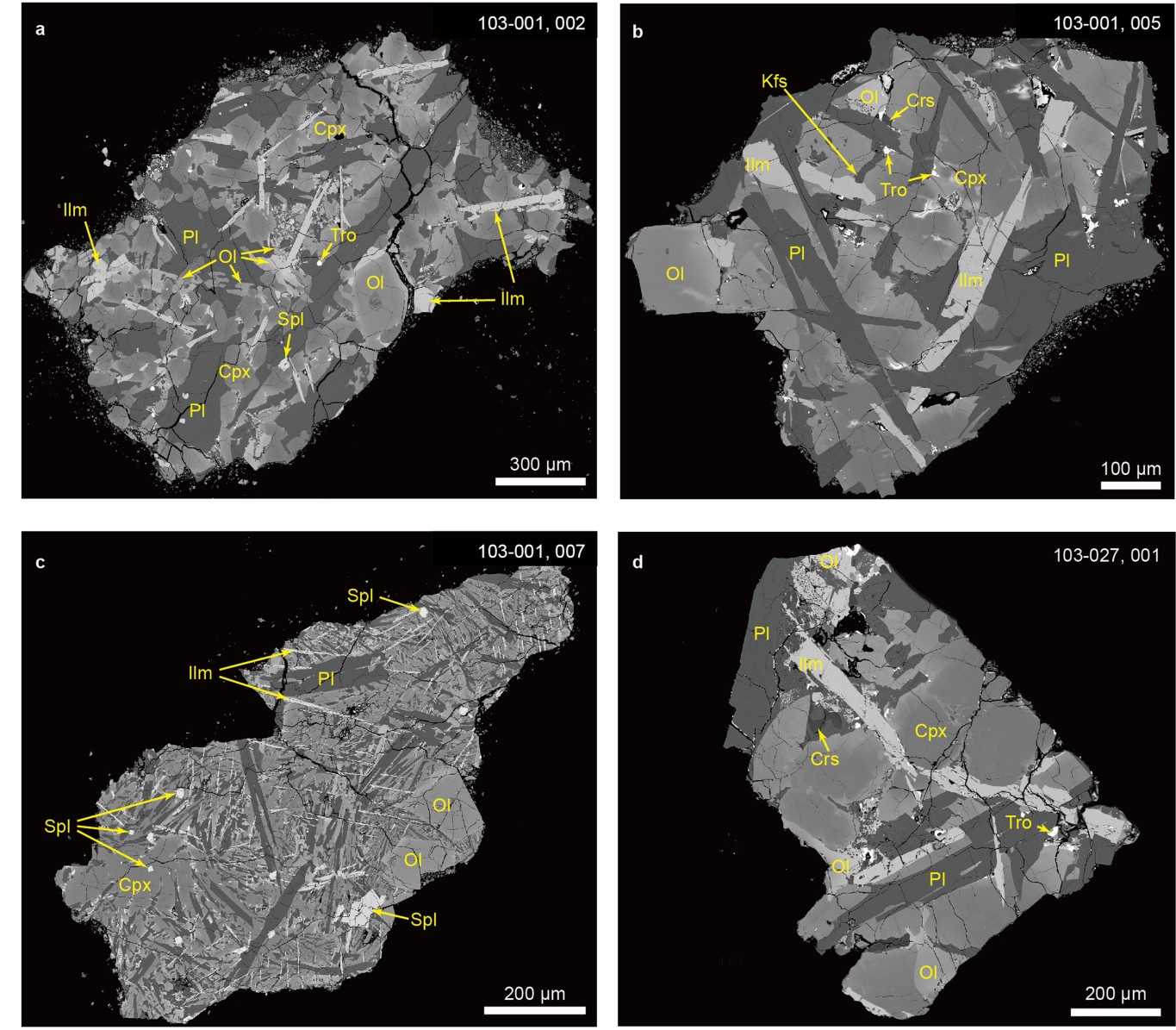

**Extended Data Fig. 1 | Representative four types of the Chang'e-5 basalt clasts. a**, Poikilitic clast. **b**, Subophitic clast. **c**, Porphyritic clast. **d**, Equigranular clast. Cpx, clinopyroxene; Pl, plagioclase; Ol, olivine; Ilm, ilmenite; Spl, spinel; Tro, troilite; Crs, cristobalite. For detailed sample description, see Methods.

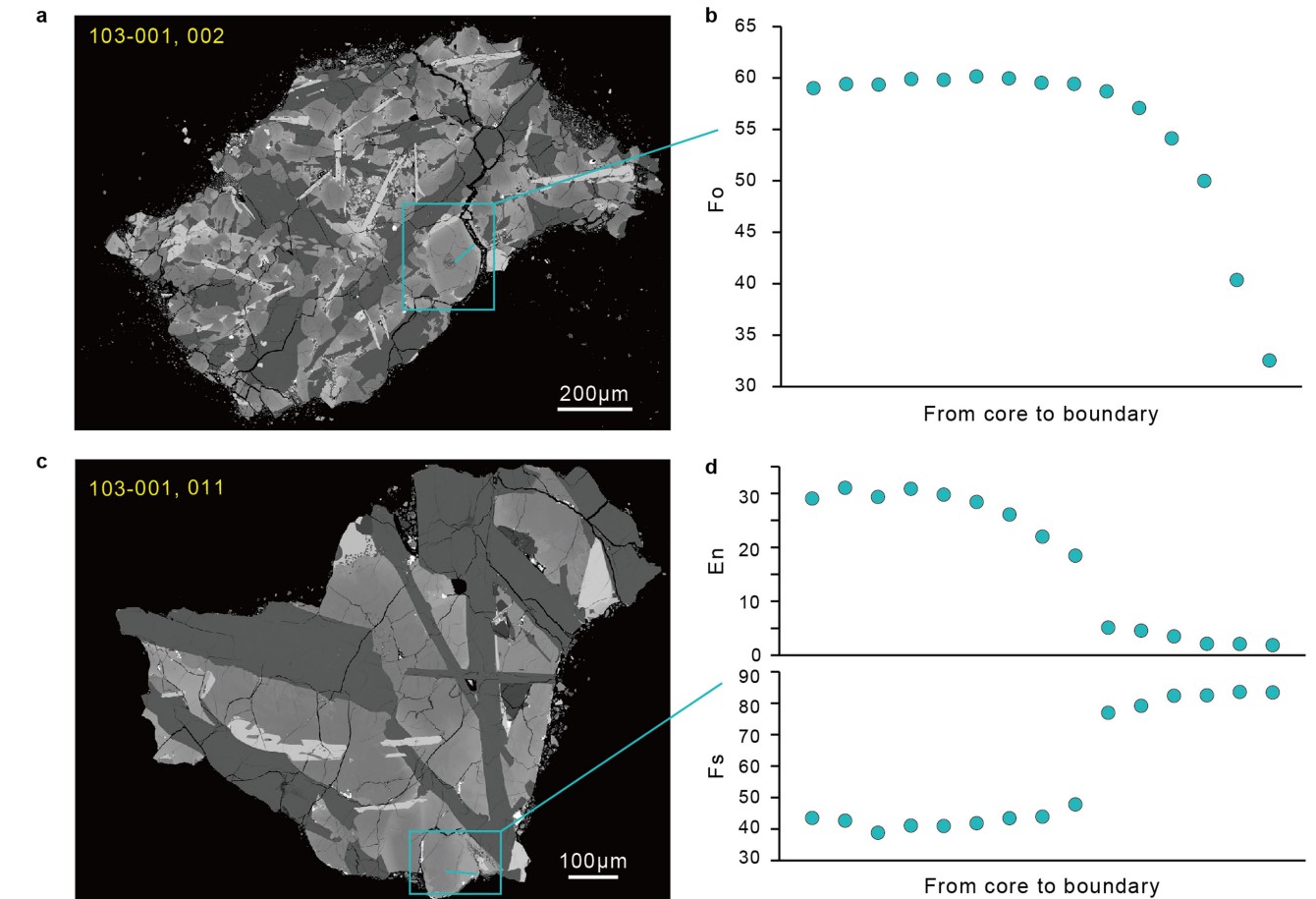

**Extended Data Fig. 2 | Compositional zoning profiles of olivine and clinopyroxene in the Chang'e-5 basalts. a**, **b**, One poikilitic clast and the variation of Fo content across an olivine grain. **c**, **d**, One subophitic clast and the variations of En and Fs contents across a clinopyroxene grain. Note that both mafic minerals have Mg-rich cores and Fe-rich rims.

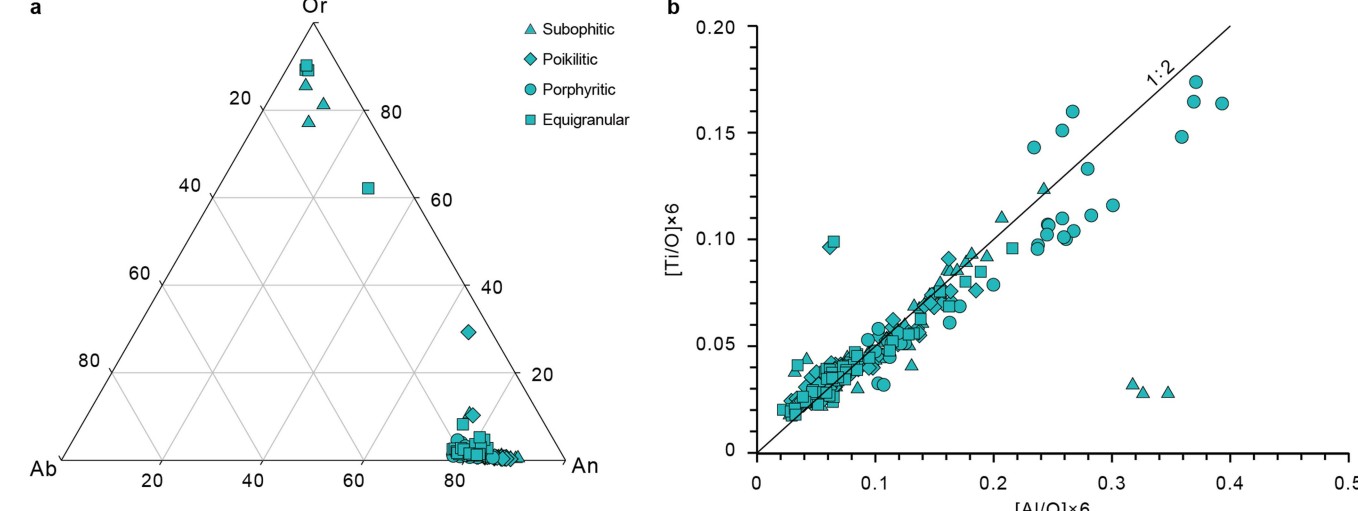

**Extended Data Fig. 3 | Compositions of feldspar and pyroxene from the Chang'e-5 basalts. a**, Ternary diagram of feldspar in the Chang'e-5 basalt clasts. Different types of basalt clast have similar plagioclase composition. K-feldspar exhibits a large compositional range. **b**, Ti versus Al diagram of the Chang'e-5 pyroxene grains. Most of the data fall along the 1:2 line.

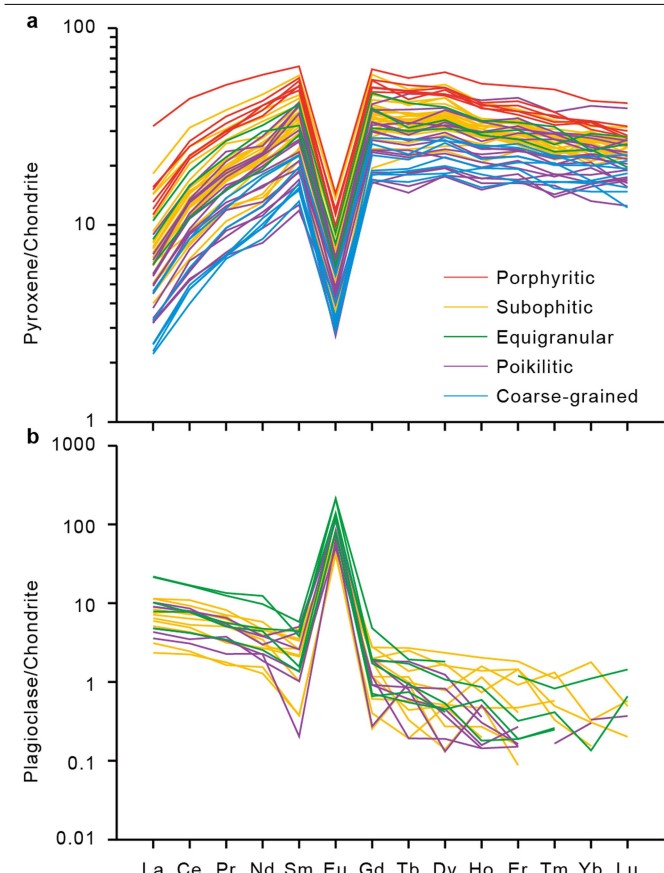

**Extended Data Fig. 4 | REE distribution of pyroxene and plagioclase from the Chang'e-5 basalts. a**, Pyroxene. **b**, Plagioclase. The normalized data are from ref. [43].

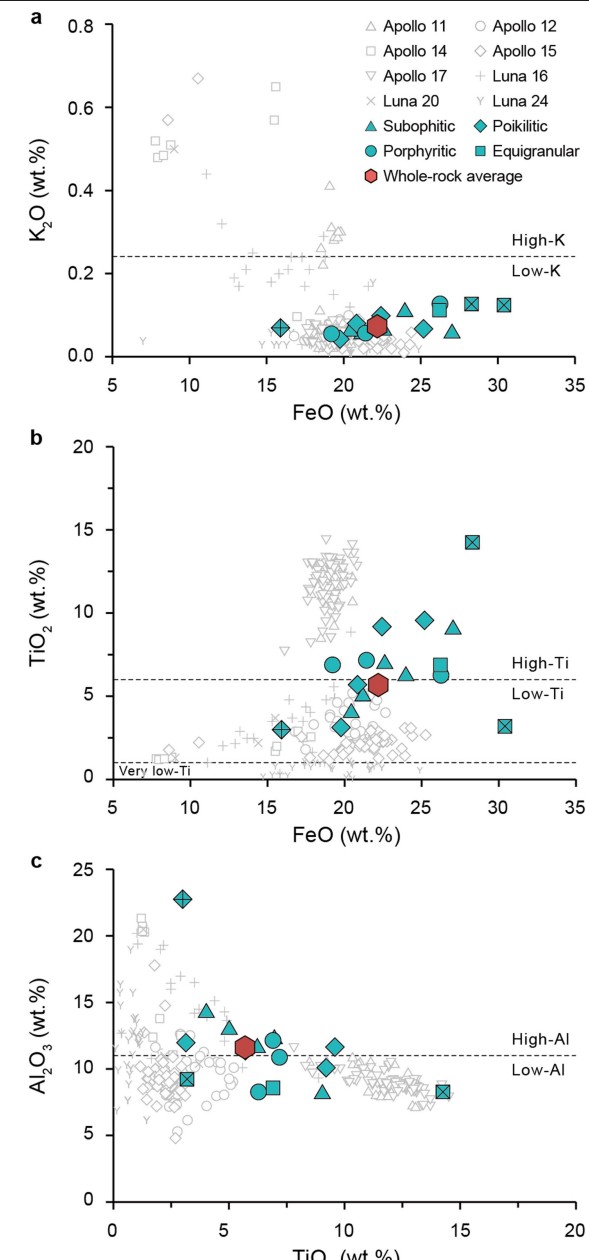

**Extended Data Fig. 5 | Compositional diagrams of estimated bulk compositions of the Chang'e-5 basalts.** The symbols of four-type clasts follow those in Extended Data Fig. 3. The red hexagonal symbols refer to the average compositions of the Chang'e-5 basalts (Extended Data Table 5). The Apollo and Luna basalts are shown for comparison (data from Clive Neal's Mare Basalt Database; https://www3.nd.edu/~cneal/Lunar-L/).

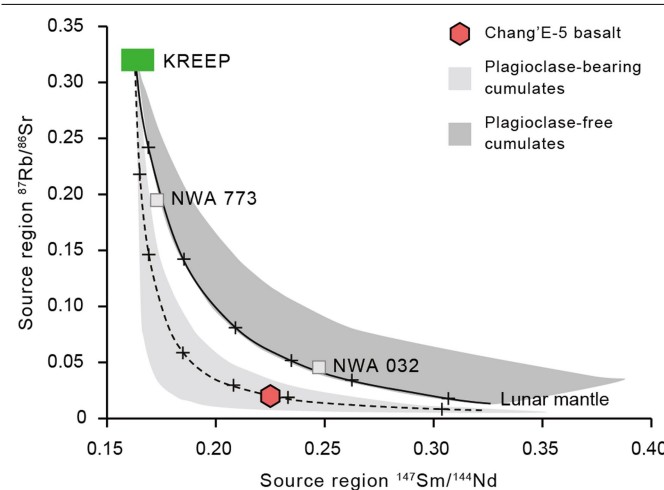

**Extended Data Fig. 6 | Calculated $^{147}Sm/^{144}Nd$ versus $^{87}Rb/^{86}Sr$ values for the source regions of the Chang'e-5 basalts.** The two grey areas, KREEP endmember, black and dashed lines (representing binary mixing between the lunar mantle and the KREEP component) are adopted from ref. [20]. The crosshair symbols represent 0.01, 0.05, 0.1, 0.2, 0.5 and 2% addition of urKREEP component. The Chang'e-5 basalt falls on the plagioclase-bearing cumulate source.

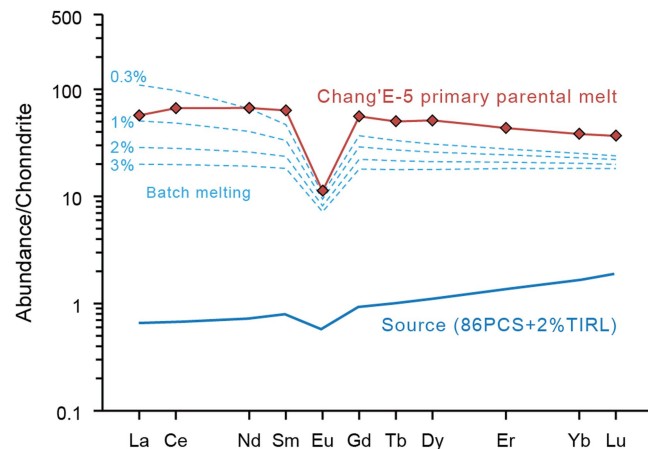

**Extended Data Fig. 7 | Chondrite-normalized REE patterns and different batch melting degrees for the calculated mantle source.** The equilibrium melt of clinopyroxene (core) with the highest MgO is chosen as the parental melt. The normalized data are from ref. [43]. PCS, percent crystallized solid; TIRL, trapped instantaneous residual liquid.

**Extended Data Table 1 | The Chang'e-5 basalt clasts analysed in this study**

| Clast No. | Size (mm) | Texture | Clinopyroxene | Olivine | Plagioclase |
|---|---|---|---|---|---|
| CE5C0000YJYX041GP* | | | | | |
| 041GP, 001 | 1.4 × 2.2 | Subophitic | $Wo_{13.6-35.9}En_{0.7-36.8}$ | | $An_{80.3-86.8}$ |
| CE5C0000YJYX042GP* | | | | | |
| 042GP, 001 | 1.9 × 3.0 | Subophitic | $Wo_{14.2-41.4}En_{1.4-26.7}$ | | $An_{75.7-86.0}$ |
| 042GP, 002‡ | 2.1 × 2.4 | Poikilitic | $Wo_{16.2-36.8}En_{21.2-45.8}$ | $Fo_{31.3-57.2}$ | $An_{81.1-89.0}$ |
| CE5C0100YJFM00103* | | | | | |
| 103-001, 002‡ | 0.9 × 1.6 | Poikilitic | $Wo_{22.8-37.3}En_{3.6-43.7}$ | $Fo_{29.7-60.1}$ | $An_{76.5-87.4}$ |
| 103-001, 003‡ | 1.4 × 1.3 | Porphyritic | $Wo_{12.6-41.7}En_{0.6-47.4}$ | | $An_{77.1-82.5}$ |
| 103-001, 005‡ | 0.6 × 0.9 | Subophitic | $Wo_{13.8-41.0}En_{2.7-31.4}$ | $Fo_{37.2-56.6}$ | $An_{81.8-85.9}$ |
| 103-001, 007‡ | 0.5 × 1.0 | Porphyritic | $Wo_{32.0-43.7}En_{21.3-36.3}$ | $Fo_{41.4-59.4}$ | $An_{77.2-85.2}$ |
| 103-001, 011‡ | 0.6 × 1.0 | Subophitic | $Wo_{11.8-37.1}En_{8.6-39.3}$ | $Fo_{37.6-48.3}$ | $An_{79.7-90.3}$ |
| 103-003, 005‡ | 0.7 × 0.8 | Poikilitic | $Wo_{15.7-39.4}En_{5.6-35.4}$ | | $An_{77.6-88.1}$ |
| 103-003, 013 | 0.6 × 0.9 | Porphyritic | $Wo_{20.5-29.0}En_{7.8-41.6}$ | $Fo_{23.3-34.9}$ | $An_{76.3-82.0}$ |
| 103-003, 015 | 1.0 × 1.0 | Fragments† | | | |
| 103-027, 001‡ | 0.7 × 0.9 | Equigranular | $Wo_{14.7-40.3}En_{3.4-33.7}$ | $Fo_{35.1-43.2}$ | $An_{80.2-82.2}$ |
| CE5C0100YJFM00406* | | | | | |
| 406-002, 002‡ | 0.7 × 1.3 | Poikilitic | $Wo_{17.1-42.8}En_{2.2-39.7}$ | $Fo_{1.5-3.4}$§ | $An_{84.1-87.8}$ |
| 406-002, 007‡ | 0.8 × 1.3 | Equigranular | $Wo_{17.5-35.0}En_{9.6-21.2}$ | $Fo_{1.0-2.2}$§ | $An_{76.3-83.1}$ |
| 406-004, 005 | 0.7 × 0.9 | Fragments† | | | |
| 406-004, 012‡ | 0.6 × 1.1 | Subophitic | $Wo_{12.6-35.1}En_{24.1-40.0}$ | $Fo_{40.7-49.8}$ | $An_{80.9-87.1}$ |
| 406-005, 002‡ | 0.4 × 0.8 | Poikilitic | $Wo_{18.5-39.1}En_{6.7-41.6}$ | $Fo_{8.9-9.5}$§ | $An_{78.1-83.9}$ |
| 406-005, 010‡ | 0.6 × 0.9 | Equigranular | $Wo_{13.7-41.1}En_{1.1-35.9}$ | | $An_{75.5-82.5}$ |

*Samples include two epoxy mounts (CE5C0000YJYX041GP and CE5C0000YJYX042GP) and two soils (CE5C0100YJFM00103 and CE5C0400YJFM00406). The basalt clasts in epoxy mounts and the soils were scooped from the lunar surface.

†Coarse-grained (>100 μm) fragments of basalt clasts, whose texture cannot be clearly identified due to the limited (2–5) grains in a single clast.

‡U-rich minerals of these clasts, for example, baddeleyite, zirconolite, tranquillityite, apatite and merrillite, were dated by the SIMS U–Pb method[9].

§The composition of olivine in mesostasis.

**Extended Data Table 2 | Modal mineralogy of the Chang'e-5 basalt clasts**

| Clast No. | Texture | Cpx | Pl | Ol | Ilm | Silica | Spl | Phos | Sul | Other |
|---|---|---|---|---|---|---|---|---|---|---|
| 041GP, 001 | Subophitic | 47.6 | 42.1 | 1.5 | 5.5 | 2.1 | | 0.7 | 0.5 | Trace |
| 042GP, 001 | Subophitic | 30.6 | 42.2 | 14.8 | 6.6 | 4.4 | 0.6 | Trace | 0.4 | Trace |
| 103-001, 005 | Subophitic | 31.6 | 42.0 | 13.9 | 8.0 | 2.9 | Trace | 0.5 | 0.4 | Trace |
| 103-001, 011 | Subophitic | 37.9 | 47.4 | 7.0 | 4.3 | 2.5 | Trace | Trace | Trace | Trace |
| 406-004, 012 | Subophitic | 39.3 | 29.3 | 17.2 | 11.3 | 0.5 | 1.0 | Trace | 0.5 | Trace |
| 042GP, 002 | Poikilitic | 33.1 | 41.7 | 19.4 | 2.9 | 1.4 | 0.7 | 0.5 | Trace | Trace |
| 103-001, 002 | Poikilitic | 37.8 | 38.9 | 13.2 | 6.2 | 2.1 | Trace | 0.4 | Trace | Trace |
| 406-002, 002 | Poikilitic | 6.5 | 72.8 | 11.8 | 3.3 | 3.3 | Trace | 1.3 | 0.9 | Trace |
| 103-003, 005 | Poikilitic | 33.2 | 41.6 | 8.9 | 11.9 | 2.1 | Trace | 1.2 | 1.0 | Trace |
| 406-005, 002 | Poikilitic | 43.3 | 37.1 | 1.4 | 11.4 | 5.4 | | 0.6 | 0.7 | Trace |
| 103-001, 003 | Porphyritic | 46.2 | 32.9 | 9.3 | 7.3 | 2.1 | 0.7 | 0.4 | 0.4 | Trace |
| 103-001-007 | Porphyritic | 42.7 | 35.2 | 13.6 | 5.8 | 1.6 | 0.9 | Trace | Trace | Trace |
| 103-003, 013 | Porphyritic | 53.0 | 28.3 | 9.0 | 7.2 | 1.5 | 0.4 | 0.4 | Trace | Trace |
| 406-002, 007 | Equigranular | 30.9 | 34.0 | 22.6 | 3.4 | 7.8 | Trace | 0.7 | 0.4 | Trace |
| 406-005, 010 | Equigranular | 46.1 | 31.8 | Trace | 19.1 | Trace | | 1.5 | 1.2 | Trace |
| 103-027, 001 | Equigranular | 49.5 | 29.9 | 7.9 | 8.1 | 3.2 | Trace | 0.5 | 0.6 | Trace |
| 103-003, 015 | Fragments | 14.2 | 81.2 | 2.6 | Trace | 1.0 | | 0.8 | | Trace |
| 406-004, 005 | Fragments | 65.6 | 17.7 | 5.7 | Trace | 1.5 | 6.6 | 2.2 | Trace | Trace |

The modes of the minerals are calculated based on SEM electron dispersive spectrometer image pixel counting using the Nanomin tool with an error of ~5%. Cpx, clinopyroxene; Pl, plagioclase; Ol, olivine; Ilm, ilmenite; Spl, spinel; Phos, phosphate; Sul, sulfide.

**Extended Data Table 3 | Rb–Sr isotopic data of plagioclase from the Chang'e-5 basalts**

| Sample | $^{87}Rb/^{86}Sr$ | 2σ | $^{87}Sr/^{86}Sr$ | 2σ | $^{87}Sr/^{86}Sr(i)$ | 2σ |
|---|---|---|---|---|---|---|
| 001-011Sr-1 | 0.00123 | 0.00002 | 0.69937 | 0.00010 | 0.69934 | 0.00010 |
| 001-011Sr-2 | 0.00124 | 0.00002 | 0.69957 | 0.00009 | 0.69954 | 0.00009 |
| 001-011Sr-3 | 0.00137 | 0.00002 | 0.69951 | 0.00009 | 0.69947 | 0.00009 |
| 001-011Sr-4 | 0.00128 | 0.00002 | 0.69948 | 0.00010 | 0.69945 | 0.00010 |
| 001-002Sr-2 | 0.00172 | 0.00004 | 0.69974 | 0.00012 | 0.69969 | 0.00012 |
| 002-002Sr-1 | 0.00145 | 0.00002 | 0.69975 | 0.00010 | 0.69970 | 0.00010 |
| 002-002Sr-2 | 0.00084 | 0.00002 | 0.69940 | 0.00010 | 0.69937 | 0.00010 |
| 002-002Sr-3 | 0.00133 | 0.00014 | 0.69964 | 0.00009 | 0.69960 | 0.00009 |
| 003-005Sr-2 | 0.00260 | 0.00009 | 0.69977 | 0.00009 | 0.69970 | 0.00009 |
| 003-015Sr-1 | 0.00146 | 0.00022 | 0.69974 | 0.00009 | 0.69970 | 0.00009 |
| 003-015Sr-2 | 0.00170 | 0.00036 | 0.69985 | 0.00010 | 0.69980 | 0.00010 |
| 004-005Sr-1 | 0.00222 | 0.00001 | 0.69992 | 0.00005 | 0.69986 | 0.00005 |
| 004-012Sr-1 | 0.00428 | 0.00010 | 0.69991 | 0.00009 | 0.69979 | 0.00009 |
| 005-002Sr-1 | 0.00490 | 0.00052 | 0.69997 | 0.00009 | 0.69982 | 0.00009 |
| 005-010Sr-1 | 0.00171 | 0.00001 | 0.69987 | 0.00008 | 0.69982 | 0.00008 |
| 005-010Sr-2 | 0.00514 | 0.00114 | 0.69969 | 0.00008 | 0.69954 | 0.00008 |
| 041GP-001Sr-1 | 0.00112 | 0.00002 | 0.69975 | 0.00011 | 0.69972 | 0.00011 |
| 042GP-002Sr-1 | 0.00095 | 0.00002 | 0.69977 | 0.00011 | 0.69974 | 0.00011 |
| 042GP-002Sr-2 | 0.00085 | 0.00002 | 0.69975 | 0.00012 | 0.69972 | 0.00012 |
| 042GP-002Sr-3 | 0.00086 | 0.00002 | 0.69952 | 0.00013 | 0.69949 | 0.00013 |
| Standard | | | | | | |
| JH56 01 | 0.00299 | 0.00011 | 0.71219 | 0.00006 | | |
| JH56 02 | 0.00143 | 0.00009 | 0.71219 | 0.00006 | | |
| JH56 03 | 0.00119 | 0.00004 | 0.71213 | 0.00006 | | |
| JH56 04 | 0.00127 | 0.00011 | 0.71203 | 0.00006 | | |
| JH56 05 | 0.00467 | 0.00039 | 0.71221 | 0.00006 | | |
| JH56 06 | 0.00452 | 0.00042 | 0.71216 | 0.00006 | | |
| JH56 07 | 0.00260 | 0.00008 | 0.71214 | 0.00006 | | |
| JH56 08 | 0.00135 | 0.00006 | 0.71219 | 0.00006 | | |
| JH56 09 | 0.00399 | 0.00044 | 0.71218 | 0.00005 | | |
| JH56 10 | 0.00447 | 0.00015 | 0.71203 | 0.00006 | | |
| JH56 11 | 0.00187 | 0.00022 | 0.71210 | 0.00005 | | |
| JH56 12 | 0.00059 | 0.00003 | 0.71228 | 0.00007 | | |
| JH56 13 | 0.00200 | 0.00018 | 0.71205 | 0.00007 | | |
| JH56 14 | 0.00182 | 0.00025 | 0.71232 | 0.00005 | | |
| JH56 15 | 0.00072 | 0.00002 | 0.71211 | 0.00006 | | |
| JH56 16 | 0.00080 | 0.00002 | 0.71219 | 0.00006 | | |
| JH56 17 | 0.00110 | 0.00005 | 0.71204 | 0.00007 | | |
| JH56 18 | 0.00143 | 0.00012 | 0.71214 | 0.00006 | | |
| JH56 19 | 0.00047 | 0.00001 | 0.71206 | 0.00005 | | |
| JH56 20 | 0.00076 | 0.00001 | 0.71229 | 0.00006 | | |
| JH56 21 | 0.00260 | 0.00026 | 0.71216 | 0.00005 | | |
| JH56 22 | 0.00074 | 0.00003 | 0.71212 | 0.00006 | | |
| JH56 23 | 0.00139 | 0.00007 | 0.71218 | 0.00006 | | |
| JH56 24 | 0.00321 | 0.00015 | 0.71221 | 0.00005 | | |
| JH56 25 | 0.00120 | 0.00002 | 0.71229 | 0.00006 | | |
| JH56 26 | 0.00172 | 0.00009 | 0.71217 | 0.00006 | | |
| JH56 27 | 0.00169 | 0.00007 | 0.71209 | 0.00007 | | |

The initial $^{87}Sr/^{86}Sr$ isotopic ratios are calculated based on the Pb/Pb age of 2.03 Ga.

**Extended Data Table 4 | Sm–Nd isotopic data of merrillite from the Chang'e-5 basalts**

| Sample | $^{147}Sm/^{144}Nd$ | $2\sigma$ | $^{143}Nd/^{144}Nd$ | $2\sigma$ | $^{145}Nd/^{144}Nd$ | $2\sigma$ | $\varepsilon_{Nd}(t)$ | $2\sigma$ |
|---|---|---|---|---|---|---|---|---|
| 004-005Nd-1 | 0.15856 | 0.00029 | 0.512532 | 0.000061 | 0.348316 | 0.000063 | 7.9 | 1.2 |
| 004-005Nd-2 | 0.15833 | 0.00029 | 0.512558 | 0.000039 | 0.348445 | 0.000050 | 8.5 | 0.8 |
| 003-015Nd-1 | 0.15696 | 0.00051 | 0.512556 | 0.000083 | 0.348488 | 0.000182 | 8.8 | 1.6 |
| 042GP-002Nd-1 | 0.15488 | 0.00031 | 0.512555 | 0.000072 | 0.348405 | 0.000082 | 9.3 | 1.4 |
| 042GP-002Nd-2 | 0.15475 | 0.00032 | 0.512536 | 0.000042 | 0.348412 | 0.000044 | 9.0 | 0.8 |
| Standard | | | | | | | | |
| SDG apatite 01 | 0.07176 | 0.00010 | 0.510937 | 0.000059 | 0.348410 | 0.000031 | | |
| SDG apatite 02 | 0.07242 | 0.00014 | 0.510946 | 0.000050 | 0.348386 | 0.000032 | | |
| SDG apatite 03 | 0.07202 | 0.00017 | 0.510924 | 0.000053 | 0.348369 | 0.000034 | | |
| SDG apatite 04 | 0.07191 | 0.00016 | 0.510865 | 0.000052 | 0.348377 | 0.000031 | | |
| SDG apatite 05 | 0.07194 | 0.00017 | 0.510943 | 0.000054 | 0.348346 | 0.000032 | | |
| SDG apatite 06 | 0.07229 | 0.00017 | 0.510908 | 0.000054 | 0.348377 | 0.000034 | | |
| SDG apatite 07 | 0.07165 | 0.00012 | 0.510930 | 0.000051 | 0.348407 | 0.000030 | | |
| SDG apatite 08 | 0.07120 | 0.00012 | 0.510891 | 0.000047 | 0.348396 | 0.000028 | | |
| LREE Glass 01 | 0.24391 | 0.00014 | 0.512148 | 0.000058 | 0.348381 | 0.000031 | | |
| LREE Glass 02 | 0.24416 | 0.00012 | 0.512099 | 0.000063 | 0.348329 | 0.000036 | | |
| LREE Glass 03 | 0.24429 | 0.00020 | 0.512075 | 0.000055 | 0.348389 | 0.000035 | | |
| LREE Glass 04 | 0.24414 | 0.00015 | 0.512109 | 0.000058 | 0.348402 | 0.000035 | | |
| LREE Glass 05 | 0.24272 | 0.00025 | 0.512063 | 0.000060 | 0.348397 | 0.000033 | | |
| LREE Glass 06 | 0.24356 | 0.00012 | 0.512095 | 0.000064 | 0.348352 | 0.000040 | | |
| LREE Glass 07 | 0.24597 | 0.00012 | 0.512113 | 0.000053 | 0.348351 | 0.000034 | | |
| LREE Glass 08 | 0.24300 | 0.00012 | 0.512095 | 0.000064 | 0.348363 | 0.000041 | | |

The $\varepsilon_{Nd}(t)$ values are calculated based on the Pb/Pb age of 2.03 Ga.

**Extended Data Table 5 | Estimated major element compositions of the Chang'e-5 basalts**

| Clast No. | Area (mm²) | SiO₂ | TiO₂ | Al₂O₃ | Cr₂O₃ | FeO | MnO | MgO | CaO | Na₂O | K₂O | P₂O₅ | Total | Mg# |
|---|---|---|---|---|---|---|---|---|---|---|---|---|---|---|
| 041GP, 001 | 2.82 | 43.7 | 5.0 | 13.0 | 0.1 | 21.2 | 0.3 | 3.2 | 12.3 | 0.6 | 0.1 | 0.3 | 99.7 | 21.4 |
| 042GP, 001 | 4.45 | 42.1 | 6.2 | 11.6 | 0.2 | 24.0 | 0.3 | 4.1 | 10.3 | 0.6 | 0.1 | 0.2 | 99.6 | 23.5 |
| 103-001, 005 | 0.49 | 40.5 | 7.0 | 12.3 | 0.1 | 22.6 | 0.3 | 5.3 | 10.8 | 0.6 | 0.1 | 0.2 | 99.7 | 29.7 |
| 103-001, 011 | 0.53 | 44.1 | 4.0 | 14.2 | 0.1 | 20.4 | 0.3 | 4.0 | 12.0 | 0.6 | 0.1 | 0.1 | 99.8 | 26.1 |
| 406-004, 012 | 0.56 | 37.5 | 9.0 | 8.1 | 0.4 | 27.0 | 0.3 | 8.2 | 8.4 | 0.4 | 0.1 | 0.1 | 99.6 | 35.3 |
| 042GP, 002 | 3.14 | 42.4 | 3.1 | 12.0 | 0.3 | 19.8 | 0.2 | 9.7 | 10.7 | 0.5 | 0.0 | 0.2 | 99.1 | 46.9 |
| 103-001, 002 | 1.52 | 42.0 | 5.7 | 11.4 | 0.2 | 20.8 | 0.3 | 7.4 | 11.2 | 0.5 | 0.1 | 0.2 | 99.8 | 39.0 |
| 406-002, 002 | 0.81 | 42.4 | 3.0 | 22.8 | 0.0 | 15.9 | 0.2 | 0.6 | 13.5 | 1.0 | 0.1 | 0.6 | 100.0 | 6.4 |
| 103-003, 005 | 0.40 | 38.3 | 9.6 | 11.7 | 0.1 | 25.2 | 0.3 | 3.1 | 10.4 | 0.6 | 0.1 | 0.5 | 99.7 | 18.1 |
| 406-005, 002 | 0.21 | 42.5 | 9.2 | 10.1 | 0.1 | 22.4 | 0.3 | 4.3 | 10.2 | 0.6 | 0.1 | 0.3 | 100.1 | 25.7 |
| 103-001, 003 | 1.60 | 41.1 | 7.2 | 10.9 | 0.4 | 21.4 | 0.3 | 7.2 | 10.7 | 0.6 | 0.1 | 0.2 | 99.8 | 37.7 |
| 103-001, 007 | 0.38 | 40.5 | 6.9 | 12.2 | 0.4 | 19.2 | 0.2 | 7.6 | 12.1 | 0.6 | 0.1 | 0.1 | 99.9 | 41.6 |
| 103-003, 013 | 0.40 | 42.2 | 6.3 | 8.3 | 0.1 | 26.2 | 0.3 | 5.5 | 9.9 | 0.5 | 0.1 | 0.2 | 99.5 | 27.4 |
| 406-002, 007 | 0.81 | 44.3 | 3.2 | 9.2 | 0.0 | 30.4 | 0.4 | 2.7 | 8.3 | 0.6 | 0.1 | 0.3 | 99.6 | 13.8 |
| 406-005, 010 | 0.41 | 34.8 | 14.3 | 8.3 | 0.1 | 28.3 | 0.3 | 2.9 | 9.4 | 0.5 | 0.1 | 0.6 | 99.4 | 15.6 |
| 103-027, 001 | 0.49 | 42.2 | 6.9 | 8.6 | 0.1 | 26.2 | 0.3 | 4.3 | 10.6 | 0.5 | 0.1 | 0.2 | 100.0 | 22.8 |
| **Weighted Average** | | **42.1** | **5.7** | **11.6** | **0.2** | **22.2** | **0.3** | **5.8** | **10.9** | **0.6** | **0.1** | **0.2** | **99.6** | **32.1** |
| 1SD | | **1.9** | **1.9** | **1.9** | **0.1** | **2.7** | **0.1** | **2.1** | **1.0** | **0.1** | **0.1** | **0.1** | **0.3** | |

The weighted average is calculated based on 13 clasts. Three outlier clasts (406-002, 002; 406-002, 007; 406-005, 010) are excluded due to their mineral abundances deviating from those of other clasts.

**Extended Data Table 6 | Estimated trace element compositions of the Chang'e-5 basalts**

| | Average composition of pyroxene* | Estimated bulk composition for Chang'E-5 basalt[†] | Estimated primary parental melt[‡] | 86PCS[§] | 86PCS+2% TIRL[§] |
|---|---|---|---|---|---|
| La | 1.7 | 37.5 | 13.42 | 0.0035 | 0.1542 |
| Ce | 7.7 | 105.2 | 40.22 | 0.0171 | 0.4041 |
| Pr | 1.6 | | | | |
| Nd | 10.2 | 66.0 | 30.25 | 0.0387 | 0.3272 |
| Sm | 4.6 | 18.2 | 9.33 | 0.0244 | 0.1176 |
| Eu | 0.4 | 1.2 | 0.63 | 0.0049 | 0.0322 |
| Gd | 6.5 | 19.2 | 10.99 | 0.0615 | 0.1825 |
| Tb | 1.1 | 3.0 | 1.82 | 0.0146 | 0.0366 |
| Dy | 8.0 | 19.6 | 12.43 | 0.1238 | 0.2681 |
| Ho | 1.6 | | | | |
| Er | 4.6 | 10.4 | 6.9 | 0.1285 | 0.2171 |
| Tm | 0.6 | | | | |
| Yb | 4.1 | 9.2 | 6.21 | 0.1923 | 0.2721 |
| Lu | 0.6 | 1.3 | 0.89 | 0.0349 | 0.0457 |
| | | | | | |
| Th | 0.1 | 4.5 | | | |
| La/Sm | | 2.1 | | | |

*Average value of the all measured pyroxenes of the Chang'e-5 clasts (Supplementary Table 2).

[†]Estimation based on the partition coefficients (Supplementary Table 3).

[‡]The clinopyroxene core with the highest MgO content (sample 406-004, 005; Supplementary Fig. 3) is chosen to calculate the parental melt. This analysed pyroxene in sample 406-004, 005 has the lowest trace element concentrations (Extended Data Fig. 4, Supplementary Table 2). We use the average data of the five measurements (406-004, 005py-1; 406-004, 005py-2; 406-004, 005py-3; 406-004, 005py-4; and 406-004, 005py-5).

[§]Data digitized from ref. [22]. The 86 PCS + 2% refers to the addition of 2% TIRL from the 86 PCS level.