## [Peer Review File · Nature]

Manuscript Title: Non-KREEP origin for Chang'E-5 basalts in the Procellarum KREEP Terrane

Reviewer Comments & Author Rebuttals

Reviewer Reports on the Initial Version:

Referee #1:

This manuscript presents some of the first data derived from the mare basalts returned by the Chang'E-5 mission. As such, it is an important manuscript that is appropriate for Nature. It furthers our understanding of the thermal and magmatic history of the Moon. The manuscript describes the texture, mineral chemistries, major and trace element chemistries, and Nd and Sr isotopic systematics of the returned basalts. Several highlights are (1) this is the youngest mare basalts thus far examined by almost 1 billion years; (2) the authors make the point that although it has a KREEP-like REE pattern, it is derived from an isotopically depleted (and non-KREEP) source. This is unlike most other basalts returned by the Apollo and Luna missions; (3) a fractional crystallization model is proposed to derive these trace element patterns (e.g., REE) and therefore the incompatible element enrichments reflect process and not source; (4) they conclude that it is somewhat unusual in that these basalts were collected from the Procellarum KREEP Terrain (PKT) and yet the source of the basalts contain such an insignificant KREEP component.

Overall, the manuscript is well written, and the data is well presented in the text and appendix. It appears that the statistical analysis of the data is correct, and in most cases that the data can statistically be used to answer some of the questions examined in the manuscript. There are alternative interpretations of these data that the authors do not consider. However, their interpretations of the data are not incorrect. The author's should address some of the following:

(1) There are young mare basalts (2.8-3.0 Ga) that have similar characteristics as the basalts reported in this manuscript (e.g., depleted source, yet enriched in incompatible elements). One in particular is lunar meteorite NWA 032. The authors of one of the NWA 032 manuscripts suggested that this basalt was produced via small degrees of partial melting and extensive fractional crystallization. The authors should at the very least acknowledge this basaltic meteorite and reference the manuscript: Borg, L. E., Gaffney, A. M., Shearer, C. K., DePaolo, D. J., Hutcheon, I. D., Owens, T. L., ... & Brennecka, G. (2009). Mechanisms for incompatible-element enrichment on the Moon deduced from the lunar basaltic meteorite Northwest Africa 032. *Geochimica et Cosmochimica Acta*, 73(13), 3963-3980.

(2) The authors state that there are a variety of textures represented by these basalts and yet the basalts most likely represent similar basaltic magmas. This is not surprising. Many of the Apollo basalts from most likely the same flow have widely different textures that are controlled by cooling rate. If it would help their discussion, a reference to Bence and Papike (1972) or Shearer et al., (1989) would make their arguments clearer.

(3) A 4420 Ma model age is given for the mantle source region of the Chang'E-5 mare basalts. Error associated with this age should be inserted into the Figure caption (Figure 2). Also, the interpretation of this age should be given. It should be compared to other model ages calculated for mantle sources for lunar magmas.

(4) The modeled melt REE composition in Figure 3b does not seem to match the Chang'E-5 REE bulk melt pattern in Figure 3a. For example, the following ratios appear to be different Ce/Sm, Sm/Eu, and Ce/Lu. Therefore, it appears that extensive fractional crystallization is not the only processes that produces the incompatible element enrichment. Perhaps, the modeling in Borg et

al. (2009) would help.

Charles "Chip" Shearer

Referee #2:

This is a very interesting paper that well illustrates the exciting new information to be gained by additional sample collection from the Moon. The important conclusion here is that the authors have shown the Chang'E basalts to have trace element characteristics similar to KREEP, but with isotope data that show clearly they cannot be derived from a KREEP source. Given that orbital compositional mapping shows much of the PKT terrane to have high K, Th, and U concentrations typical of KREEP, the Chang'E results reported here suggest that evolved basalts, and not just KREEP, contribute to this signal. As the authors conclude, this may mean that KREEP is not as widely distributed as previously thought, with consequences for the chemical structure and thermal evolution of the lunar interior.

The paper is well written and presents a large quantity of data, most of which is in supplemental files. The data appear to be of high quality. The one obvious criticism of the results is that the very small size of the samples makes it difficult to obtain accurate whole rock compositions. The authors properly acknowledge this problem, but at least some of the calculated whole rock compositions (e.g., those with SiO₂ below 40%) likely reflect modal mineralogies in the small samples that don't well reflect the true modal mineralogy of the basalt the chips came from. As the authors don't make detailed comparisons of their major element data with the data from Apollo samples where sampling size was not a problem, I don't consider this a major shortcoming of the paper.

I will point out, however, that taking a weighted average of the composition of small chips from a basalt will not necessarily give a better estimate of the bulk composition of the basalt. For example, in figure 1b, I will guess that the samples with Mg# below 20 may reflect chips with too much ilmenite and fayalitic olivine compared to the true whole rock. If so, leaving out those samples from the average might push the true composition of the Chang'E basalt closer to the Apollo 12 range.

There is not a lot the authors can do about this problem given the samples available, but they may want to be a bit more cautious in their claims that the Chang'E basalts are compositionally unlike anything seen previously on the Moon, though in age, they certainly are novel. For example, many Luna 16 and 20 analyses also extend to quite evolved compositions. A similar criticism can be raised about figure 3 as, for example, Apollo 14 basalts show chondrite normalized La abundances that range from 10 to about 80 (e.g. Shervais et al., 1985), so illustrating the Apollo data using a single pattern for each site is a bit misleading.

One other point is that some aspects of the discussion depend on data that is reported in reference 9, which is not yet published. This reference provides the age, which is critical to the paper, and a commonly mentioned U/Pb ratio that is useful, but not essential, to the discussion in the paper. In this case, I do not think the unpublished nature of reference 9 should hold up consideration of this paper.

Overall, this is an excellent presentation of high-quality data on an important subject that uses a unique sample set. The paper should be published in Nature with only minimal revisions needed. I've listed a number of comments on the details below, but none of these should require significant revisions to address. I congratulate the authors on the mission and the exciting data they were able to obtain from these samples.

Rick Carlson 8/12/2021

Detailed comments keyed to line numbers:

Line 29: Many Luna basalts are similarly highly evolved.

Line 32: An important conclusion here, but it also may be worth noting that the degrees of melting required to produce these basalts is very low, which would be consistent with a relatively cool lunar mantle by this time.

Line 39: I'm confused by the wording here. Do you mean "...from the two soil samples that were composed of ~45% basalt."

Line 45: Can you say which minerals were dated? Are these Pb-Pb isochrons or for minerals with extremely radiogenic Pb?

Line 96, 97: Actually, both Luna 16 and 20 samples get down to Mg# of 30 and Luna 16 basalts have about 5% TiO₂, so similar to the basalts analyzed here.

Line 99: Should be "high LREE enrichment" or "and are highly LREE enriched"

Line 116: There is a lot of dependence on unpublished reference 9, but none that is critical to the discussion in this paper.

Line 129: "crystallization" not "crystallization", similar problem in the figure 3 caption. Perhaps "s" is the British English spelling? If so, then using "z" in the supplemental file is incorrect. Please be consistent in the spelling.

Line 132: Many of the Luna samples are also quite highly evolved, so the idea that at least some mare basalts experienced extensive fractional crystallization is not new, but this explanation does seem particularly relevant for the basalts analyzed here.

Line 159: The authors suggest that these basalts are the result of quite low degrees of partial melting, so the lunar interior likely was substantially cooler at 2 Gyr than at ~3.5 Gyr when the variety of more primitive basalts sampled by Apollo were formed.

Line 315: I am very impressed by the quality of the Sr and Nd isotope data produced by this technique, so please don't take the following comment as a criticism, but I've always wondered why the laser ablation community doesn't measure 143/146 ratios fractionation corrected using 145/146, this way no Sm correction is needed. Yes, 145 is a small isotope only a mass unit away from 146, but I suspect that the number of ions counted dominates the precision, not the relatively small 145/146 ratio. The authors don't need to respond to this comment.

Supplemental file:

Line 21: "scooped" not "scrooped"

Line 34: "dominate" not "dominated"

Line 58: "compositionally" not "compositional"

Line 68: "significant" not "significantly"

Author Rebuttals to Initial Comments:

Referee #1, Charles Shearer:

There are alternative interpretations of these data that the authors do not consider. However, their interpretations of the data are not incorrect. The author's should address some of the following:

(1) There are young mare basalts (2.8-3.0 Ga) that have similar characteristics as the basalts reported in this manuscript (e.g., depleted source, yet enriched in incompatible elements). One in particular is lunar meteorite NWA 032. The authors of one of the NWA 032 manuscripts suggested that this basalt was produced via small degrees of partial melting and extensive fractional crystallization. The authors should at the very least acknowledge this basaltic meteorite and reference the manuscript: Borg, L. E., Gaffney, A. M., Shearer, C. K., DePaolo, D. J., Hutcheon, I. D., Owens, T. L., ... & Brennecka, G. (2009). Mechanisms for incompatible-element enrichment on the Moon deduced from the lunar basaltic meteorite Northwest Africa 032. *Geochimica et Cosmochimica Acta*, 73(13), 3963-3980.

Response: We agree with this suggestion. Some lunar meteorites (NWA 032, NWA 4734, LAP 02205) have similar characteristics (depleted Sr-Nd isotopes with enriched incompatible elements) and are suggested to be derived from low-degree partial melting of a depleted source (Borg et al., 2009; Elardo et al., 2014). These previous studies are beneficial for us to understand the origin of the Chang'E-5 basalt. Our modelling is consistent with the modeling in Borg et al. (2009). For example, the source of Chang'E-5 basalt is plagioclase-bearing (Extended Data Fig 6) as indicated by the modeling in Borg et al. (2009). Our modeling also assumes the source contains plagioclase. The reason why low-degree (2-3%) partial melting and extensive (43-78%) fractional crystallisation are both required in our modelling is that the Chang'E-5 basalt has higher incompatible trace element concentrations than NWA 032 (e.g., La = 37.5 ppm vs. 11.0~12.3 ppm). Partial melting alone requires an unrealistically low degree (<0.3%) and cannot simultaneously reproduce the LREE and HREE contents in the Chang'E-5 basalt. The compositional zoning in olivine and pyroxene also support our interpretation. Thus, we acknowledge the previous studies (lines 99 to 100) and modified Extended Data Figure 6 following the modeling in Borg et al. (2009) in the revision.

(2) The authors state that there are a variety of textures represented by these basalts and yet the basalts most likely represent similar basaltic magmas. This is not surprising. Many of the Apollo basalts from most likely the same flow have widely different textures that are controlled by cooling rate. If it would help their discussion, a reference to Bence and Papike (1972) or Shearer et al., (1989) would make their arguments clearer.

Response: Thanks for this suggestion, which is very helpful. We plot the pyroxene analyses on the Ti/Al diagram of Bence and Papike (1972). Most pyroxene grains of the Chang'E-5 basalt clasts fall along the 1:2 line (Extended Data Fig. 3b), reflecting near-simultaneous crystallization of augite, plagioclase, and ilmenite in one cooling event. This mineralogy thus provides strong evidence that these basalt clasts are from a single basaltic lava flow. In addition, the pyroxene grains measured in different clasts display similar REE patterns, which is consistent with the previous observation in Apollo 12 and Apollo 15 basalts (Shearer et al., 1989). Thus, we revised the text (lines 65 to 69) appending the suggested reference to Bence and Papike (1972) and added Extended Data Figure 3b with the suggested Ti/Al diagram.

(3) A 4420 Ma model age is given for the mantle source region of the Chang'E-5 mare basalts. Error associated with this age should be inserted into the Figure caption (Figure 2). Also, the interpretation

of this age should be given. It should be compared to other model ages calculated for mantle sources for lunar magmas.

Response: We insert the error and interpretation of this model age into the figure legend (Figure 3 in the revision). This model age may represent the timing of urKREEP formation (Nyquist and Shih, 1992). Following previous studies (Borg et al., 2009; Elardo et al., 2014), we also assume here that this 4420 Ma age could represent the formation age of the mare mantle source region, although different ages have been suggested by other models (e.g., 4389 ± 45 Ma, 4353 ± 37 Ma; Borg et al., 2015 and references therein). These ages are indistinguishable within uncertainty and have a minimal shift on the calculated $^{147}\text{Sm}/^{144}\text{Nd}$ ratio (<0.002 change for a 100 million year variation in age). Thus, while the revised text now addresses the error and interpretation of this model age as requested by the reviewer, it does not affect the conclusion of this paper.

(4) The modeled melt REE composition in Figure 3b does not seem to match the Chang'E-5 REE bulk melt pattern in Figure 3a. For example, the following ratios appear to be different Ce/Sm, Sm/Eu, and Ce/Lu. Therefore, it appears that extensive fractional crystallization is not the only processes that produces the incompatible element enrichment. Perhaps, the modeling in Borg et al. (2009) would help.

Response: We apologize for the inappropriate presentation of the data in Figure 3 (Figure 4 in the revision). The red dots in panel a and panel b were labeled "Chang'E-5 bulk composition" and "Chang'E-5 parental melt", respectively. This may have led to misunderstanding. To match the Chang'E-5 bulk composition with the high Ce/Sm, Sm/Eu, and Ce/Lu ratios, up to 78-88% fractional crystallisation is needed. To avoid confusion, we modified this figure using different symbols to distinguish "Chang'E-5 bulk composition" and "Chang'E-5 parental melt".

We agree that extensive fractional crystallization is not the only process that can produce incompatible element enrichment. As discussed above, both low-degree (2-3%) partial melting and extensive (43-78%) fractional crystallisation are required to produce the Chang'E-5 basalts. We revise the text to emphasize that both processes are necessary (lines 27 and 122).

Referee #2, Richard Carlson:

The paper is well written and presents a large quantity of data, most of which is in supplemental files. The data appear to be of high quality. The one obvious criticism of the results is that the very small size of the samples makes it difficult to obtain accurate whole rock compositions. The authors properly acknowledge this problem, but at least some of the calculated whole rock compositions (e.g., those with SiO₂ below 40%) likely reflect modal mineralogies in the small samples that don't well reflect the true modal mineralogy of the basalt the chips came from. As the authors don't make detailed comparisons of their major element data with the data from Apollo samples where sampling size was not a problem, I don't consider this a major shortcoming of the paper.

I will point out, however, that taking a weighted average of the composition of small chips from a basalt will not necessarily give a better estimate of the bulk composition of the basalt. For example, in figure 1b, I will guess that the samples with Mg# below 20 may reflect chips with too much ilmenite and fayalitic olivine compared to the true whole rock. If so, leaving out those samples from the average might push the true composition of the Chang'E basalt closer to the Apollo 12 range.

Response: We agree with this wise suggestion and carefully checked the modal mineralogy of all the clasts. Three clasts (406-002, 002; 406-002, 007; 406-005, 010) have mineral abundances that deviate from those of other clasts. Clast 406-002, 002 has an extremely high abundance of plagioclase (72.8%). Clast 406-002, 007 has very high abundances of fayalitic olivine (22.6%) and silica (7.8%). Clast 406-002, 002 has an extremely high abundance of ilmenite (19.1%). In the revision, we exclude these outlier samples from the estimation of the bulk composition (lines 377 to 382). The new results are indeed closer to the Apollo 12 and Luna 16 range.

There is not a lot the authors can do about this problem given the samples available, but they may want to be a bit more cautious in their claims that the Chang'E basalts are compositionally unlike anything seen previously on the Moon, though in age, they certainly are novel. For example, many Luna 16 and 20 analyses also extend to quite evolved compositions. A similar criticism can be raised about figure 3 as, for example, Apollo 14 basalts show chondrite normalized La abundances that range from 10 to about 80 (e.g. Shervais et al., 1985), so illustrating the Apollo data using a single pattern for each site is a bit misleading.

Response: We have deleted all claims that such basalts have never before been reported. The Chang'E-5 basalt is compositionally close to some Luna 16 and Apollo 12 basalts, although it exhibits higher FeO, TiO₂, and incompatible trace element concentrations than Luna 16 and Apollo 12 basalts.

To keep the original Figure 3 concise and straightforward, we had used a single pattern for all Apollo sites. This had concealed the variability of rare earth elements among basalts from different sampling sites. Thus, we have modified this figure (Figure 4 in revision) using two to three different patterns to present the compositional variation of the main groups of basalts in each site.

One other point is that some aspects of the discussion depend on data that is reported in reference 9, which is not yet published. This reference provides the age, which is critical to the paper, and a commonly mentioned U/Pb ratio that is useful, but not essential, to the discussion in the paper. In this case, I do not think the unpublished nature of reference 9 should hold up consideration of this paper.

Response: We agree that the age data of reference 9 are critical to this paper, thus it is retained and those authors plan to submit their revisions shortly.

Detailed comments keyed to line numbers:

Line 29: Many Luna basalts are similarly highly evolved.

Response: We have deleted all claims that such basalts have never before been reported.

Line 32: An important conclusion here, but it also may be worth noting that the degrees of melting required to produce these basalts is very low, which would be consistent with a relatively cool lunar mantle by this time.

Response: We agree with this comment. However, the summary paragraph is limited to be less than 200 words; thus, we emphasized this point in the last paragraph (lines 127 to 129).

Line 39: I'm confused by the wording here. Do you mean "...from the two soil samples that were composed of ~45% basalt."

Response: Fixed. Approximately 45% of lithic clasts are basalt.

Line 45: Can you say which minerals were dated? Are these Pb-Pb isochrons or for minerals with extremely radiogenic Pb?

Response: Minerals with extremely radiogenic Pb (baddeleyite, zirconolite, tranquillityite) of thirteen clasts were dated by *in situ* Pb-Pb method, yielding a crystallisation age of 2030 ± 4 million years ago. We revised the text to specify the minerals used (lines 41 to 42).

Line 96, 97: Actually, both Luna 16 and 20 samples get down to Mg# of 30 and Luna 16 basalts have about 5% TiO₂, so similar to the basalts analyzed here.

Response: We have deleted all claims that such basalts have never before been reported.

Line 99: Should be "high LREE enrichment" or "and are highly LREE enriched"

Response: Fixed.

Line 116: There is a lot of dependence on unpublished reference 9, but none that is critical to the discussion in this paper.

Response: We agree with this suggestion and delete the citation of the U/Pb ratios from Reference 9 here.

Line 129: "crystallisation" not "crystallization", similar problem in the figure 3 caption. Perhaps "s" is the British English spelling? If so, then using "z" in the supplemental file is incorrect. Please be consistent in the spelling.

Response: Fixed.

Line 132: Many of the Luna samples are also quite highly evolved, so the idea that at least some mare basalts experienced extensive fractional crystallization is not new, but this explanation does seem particularly relevant for the basalts analyzed here.

Response: We have deleted all claims that such basalts have never before been reported.

Line 159: The authors suggest that these basalts are the result of quite low degrees of partial melting, so the lunar interior likely was substantially cooler at 2 Gyr than at ~3.5 Gyr when the variety of more primitive basalts sampled by Apollo were formed.

Response: We agree with this nice summary by the reviewer and have add a sentence to that effect here (lines 127 to 129): “The highly evolved origin of the 2-billion-year-old Chang’E-5 basalts implies that the lunar interior was substantially cooler at that time than at ~3.5 billion years ago when the variety of more primitive basalts sampled by Apollo were formed.”

Line 315: I am very impressed by the quality of the Sr and Nd isotope data produced by this technique, so please don't take the following comment as a criticism, but I've always wondered why the laser ablation community doesn't measure $^{143}\text{Nd}/^{144}\text{Nd}$ ratios fractionation corrected using $^{145}/^{146}$, this way no Sm correction is needed. Yes, 145 is a small isotope only a mass unit away from 146, but I suspect that the number of ions counted dominates the precision, not the relatively small $^{145}/^{146}$ ratio. The authors don't need to respond to this comment.

Response: We appreciate the constructive comment. We think this divide has historical origins. Before the advent of LA-MC-ICPMS, all Nd isotopic data were obtained by solution measurement via TIMS, which are reported as $^{143}\text{Nd}/^{144}\text{Nd}$ ratios and normalized by the $^{146}\text{Nd}/^{144}\text{Nd}$ ratio. All the standards used during LA analysis are certified by solution measurement. To make comparisons simply, the LA Nd isotopic data are reported in the same format as those of the solution measurement. In reality, the final Nd isotopic results (after, in most situations, correction of isobaric interference of ^{144}Sm on the ^{144}Nd and normalization by using interference-corrected $^{146}\text{Nd}/^{144}\text{Nd}$ ratio) are comparable to those obtained by solution measurement. Therefore, it is not imperative to change the reported Nd isotopic ratio from $^{143}\text{Nd}/^{144}\text{Nd}$ to $^{143}\text{Nd}/^{146}\text{Nd}$.

Line 21: "scooped" not "scrooped"

Response: Fixed.

Line 34: "dominate" not "dominated"

Response: Fixed.

Line 58: "compositionally" not "compositional"

Response: Fixed.

Line 68: "significant" not "significantly"

Response: Fixed.

References:

- Bence, A., Papike, J., 1972. Pyroxenes as recorders of lunar basalt petrogenesis: Chemical trends due to crystal-liquid interaction, Lunar and Planetary Science Conference Proceedings, pp. 431.
- Borg, L.E., Gaffney, A.M., Shearer, C.K., 2015. A review of lunar chronology revealing a preponderance of 4.34–4.37 Ga ages. *Meteoritics & Planetary Science*, 50(4): 715-732.
- Borg, L.E., Gaffney, A.M., Shearer, C.K., DePaolo, D.J., Hutcheon, I.D., Owens, T.L., Ramon, E., Brennecke, G., 2009. Mechanisms for incompatible-element enrichment on the Moon deduced from the lunar basaltic meteorite Northwest Africa 032. *Geochimica et Cosmochimica Acta*, 73(13): 3963-3980.
- Elardo, S.M., Shearer Jr., C.K., Fagan, A.L., Borg, L.E., Gaffney, A.M., Burger, P.V., Neal, C.R., Fernandes, V.A., McCubbin, F.M., 2014. The origin of young mare basalts inferred from lunar meteorites Northwest Africa 4734, 032, and LaPaz Icefield 02205. *Meteoritics & Planetary Science*, 49(2): 261-291.

- Nyquist, L.E., Shih, C.Y., 1992. The isotopic record of lunar volcanism. *Geochimica et Cosmochimica Acta*, 56(6): 2213-2234.
- Shearer, C.K., Papike, J.J., Simon, S.B., Shimizu, N., 1989. An ion microprobe study of the intra-crystalline behavior of REE and selected trace elements in pyroxene from mare basalts with different cooling and crystallization histories. *Geochimica et Cosmochimica Acta*, 53(5): 1041-1054.

Reviewer Reports on the First Revision:

Referee #1:

As I had mentioned in my previous review, this is an important manuscript that should be published. Since that time the authors responded to my review in a reasonable manner. I do have some additional small edits that escaped my first review.

First in Table 1 for the La/Sm ratios, the authors should insert the ratio for chondrite normalized values and state that these are normalized values in the table. I think this will give the readers a clear sense of the slope of the REE pattern. Further, in this table, average analyses of Apollo samples from each site are given. However, this ignores the fact that there are different rock types at each site. In the table it would be better to compare to specific mare basalt types from the Apollo sites. Not all mare basalt types need to be given in this table.

Second, there should be a slight expansion of the details on the calculation of bulk analyses from mineral data. For example, the mesostasis and accessory phases (e.g., apatite) commonly contain a high % of incompatible elements and therefore define the bulk incompatible element content of the rock. High errors may be associated with these very heterogeneous components. Also, averaging core and rim analyses of phases (e.g., pyroxene) does not necessarily give an average pyroxene value. Therefore, a little more information on how these potential problems were remedied should be included in this presentation. Again, I think only 3-4 sentences are needed.

Referee #2:

The changes made by the authors in response to the reviewer comments very well address the minor criticisms raised in the first review. I consider the paper now ready for acceptance by Nature.

Author Rebuttals to First Revision:

Remarks to Referee #1

In the revision, we made two changes according to the comments of referee #1. First, we modified Table 1, listing five types of low Ti Apollo basalts. In addition, we added a paragraph to describe the estimation method of the bulk trace elements of the Chang'E-5 basalt (lines 385 to 390). Although the estimates may have relatively large uncertainties, because we do not use them for REE modeling, this does not affect our conclusion.